# LEARNING TO EXTRAPOLATE AND ADJUST: TWO-STAGE META-LEARNING FOR CONCEPT DRIFT IN ONLINE TIME SERIES FORECASTING

## ABSTRACT

The non-stationary nature of time series data in many real-world applications makes accurate time series forecasting challenging. In this paper, we consider concept drift where the underlying distribution or environment of time series changes. We first classify concepts into two categories, macro-drift corresponding to stable and long-term changes and micro-drift referring to sudden or short-term changes. Next, we propose a unified meta-learning framework called LEAF (**L**earning to **E**xtrapolate and **A**djust for **F**orecasting). Specifically, an extrapolation module is first meta-learnt to track the dynamics of the prediction model in latent space and extrapolate to the future considering macro-drift. Then an adjustment module incorporates meta-learnable surrogate loss to capture sample-specific micro-drift patterns. Through this two-stage framework, different types of concept drifts can be handled. In particular, LEAF is model-agnostic and can be applied to any deep prediction model. To further advance the research of concept drift on time series, we open source three electric load time series datasets collected from real-world scenarios, which exhibit diverse and typical concept drifts and are ideal benchmark datasets for further research. Extensive experiments on multiple datasets demonstrate the effectiveness of LEAF[1].

## 1 INTRODUCTION

Accurate time series forecasting (Hyndman & Athanasopoulos, 2018; Benidis et al., 2022) is of great importance in many domains, such as stock market (Cavalcante & Oliveira, 2014; Shahi et al., 2020), weather (Bi et al., 2023), energy (Yang et al., 2023; Hong et al., 2020), etc. In many real-world applications, time series data arrives as a stream. It is observed that the non-stationary nature of the times series data in many scenarios makes the model trained on the historical data outdated and leads to unsatisfying forecasting on the new data (Liu et al., 2023b; Pham et al., 2023). Thus, there is a growing interest in online time series forecasting, where the deployed model can swiftly adapt to non-stationary environments.

Formally, in this paper we study the *concept drift* problem in time series forecasting, which refers to the changing underlying distributions or environments. There are many classifications of concept drifts in times series. For example, in (Liu et al., 2023b), the concept drift is categorized into two types: real concept drift and virtual concept drift. Other concept drift groups include sudden concept drift, incremental concept drift, gradual concept drift, and recurring concept drift. In this paper, we divide it into two distinct categories, macro-drift and micro-drift, as illustrated in Figure 1. Specifically, *macro-drift* refers to stable and long-term changes in the characteristics or patterns of time series, which can be identified and characterized by analyzing a significant period of data. For instance, the electric power load time series exhibits a growth trend influenced by factors such as population growth and economic development. On the other hand, *micro-drift* refers to sudden or short-term changes in time series. Data exhibiting micro-drift may deviate from the current distribution. An example is the noticeable increase in traffic volume time series during rainy days and holidays.

Currently, the research of concept drift in time series faces several challenges. The first challenge is the lack of benchmark datasets for concept drifts. Good benchmark datasets play an importance role

---

[1]Code and data are available at `https://anonymous.4open.science/r/LEAF-66C4`

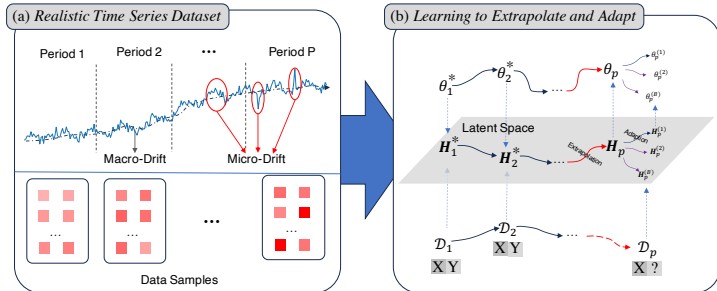

Figure 1: (a) Illustration of Macro- and Micro-Drift of Time Series. (b) Design Concept of LEAF.

in the advancements of the field, such as ImageNet (Deng et al., 2009) for the CV field. Unfortunately, the field of time series analysis field lacks benchmark datasets with high quality. Although some benchmark datasets are proposed for different time series tasks, most of them are challenged for their quality (Hahn et al., 2023). On the other hand, as pointed in (Hahn et al., 2023), the non-stationary time series benchmark datasets are missing in the current time series benchmark datasets, which hinders the development of forecasting algorithms for concept drift. The second challenge lies in the difficulty of handling different types of drifts in forecasting algorithms. The presence of complex macro- and micro-drift, as well as their potential superposition, makes capturing the evolving patterns in time series data challenging. These drifts frequently occur in real-world scenarios, making it imperative for our models to possess the capability to comprehend the concept drift phenomenon and further effectively infer the future environment.

To address the concept drift problem in online time series forecasting, we propose a unified meta-learning framework called LEAF (**L**earning to **E**xtrapolate and **A**djust for **F**orecasting). LEAF is model-agnostic and can be applied to any deep prediction model, making it highly versatile. It learns a low-dimensional latent embedding of parameters of the target prediction model, allowing it to effectively track the dynamics of the model in response to concept drift in the time series. Optimization in the low-dimensional space conforms to the characteristics of concept drift and alleviates the over-fitting issue. Specifically, LEAF consists of two stages: extrapolation and adjustment, to deal with macro- and micro-drift, respectively. In the extrapolation stage, a meta-learnt extrapolation network is employed to utilize the optimal latent embeddings from past periods and infer the latent embedding for the next period. This enables LEAF to capture and anticipate the macro-drift. In the adjustment stage, to handle micro-drift with diverse types and extensions, we introduce a meta-learnable surrogate loss for each sample, which guides the adjustment of inferred latent embedding above to become sample-specific ones. This empowers the model to capture micro-drift patterns accurately. By learning to extrapolate and adjust, the proposed LEAF framework offers a promising solution to overcome the challenges posed by concept drift in online time series forecasting, providing improved forecasting accuracy in dynamic and evolving environments.

Moreover, for the problem of limited time series benchmark datasets for concept drift, we collect and open source three real-world electric load time series datasets. These datasets exhibit typical and diverse types of concept drift, making them ideal to evaluate and compare different forecasting algorithms. To validate the effectiveness of LEAF, we conduct extensive experiments on the three electric load datasets as well as existing public benchmark datasets. Specifically, we apply LEAF to various types of deep prediction models, such as MLP, RNN, CNN, and Transformer-based models. Through evaluating LEAF's performance of different prediction models on datasets from various domains, we establish the universality and effectiveness of LEAF and highlight its robustness and applicability in addressing concept drift in online time series forecasting.

To summarize, our contributions to this work are threefold:
- We propose the LEAF framework, a unified model-agnostic meta-learning framework that effectively addresses the challenges posed by concept drift in online time series forecasting. It learns to extrapolate and adjust the target prediction model in latent space and handle macro- and micro-drifts effectively.
- We open source three electric load time series datasets collected from real-world scenarios. These datasets exhibit diverse and typical concept drifts and are ideal benchmark datasets for the study of concept drift. We believe that the real and challenging datasets can advance and promote the research of concept drift in time series.
- We conduct extensive experiments on multiple public and real-world time series benchmark datasets, demonstrating the superior performance of LEAF compared to existing approaches in various dynamic and evolving environments.

## 2 RELATED WORK

### 2.1 DEEP TIME SERIES FORECASTING

In recent years, deep learning-based forecasting methods have gained prominence within the time series forecasting landscape. Models such as FC-LSTM (Sutskever et al., 2014) combine LSTM networks with fully-connected layers to make predictions. TCN (Bai et al., 2018) introduces causal and dilated convolutions, providing a framework to model temporal patterns effectively. DeepAR (Salinas et al., 2020), on the other hand, is an LSTM-based model that predicts the parameters of future value distributions. Most notably, Transformer-based models have surged to the forefront (Wen et al., 2023). These models leverage attention mechanisms to capture long-term temporal dependencies efficiently. For example, LogTrans (Li et al., 2019) introduces local/log-sparse attention, while Reformer (Kitaev et al., 2020) approximates attention with locality-sensitive hashing. Informer (Zhou et al., 2021) employs sparsity queries for attention, and Autoformer (Wu et al., 2021) introduces series decomposition and time delay aggregation for improved periodicity modeling. FEDformer (Zhou et al., 2022) proposed that time series data can be effectively represented in the frequency domain, leading to the development of a frequency-enhanced Transformer with $O(L)$ complexity. Quatformer (Chen et al., 2022) took a novel approach by introducing learning-to-rotate attention (LRA) based on quaternions, infusing models with learnable period and phase information to capture intricate periodical patterns. Some recent developments include DLinear (Zeng et al., 2023), PatchTST (Nie et al., 2023), etc. Unfortunately, all these deep learning algorithms cannot handle concept drift quite well. Our proposed LEAF is a unified model-agnostic meta-learning framework resolving concept drift problems and can be applied to all deep prediction models mentioned above, which is confirmed by our experiments on MLP, RNN, CNN, and Transformer-based models on multiple datasets.

### 2.2 CONCEPT DRIFT AND ONLINE LEARNING

Addressing concept drift in online time series forecasting requires innovative strategies. A common approach involves optimization-based meta-learning or model-based meta-learning (Huisman et al., 2021). For example, DeepTime (Woo et al., 2022) treats different lookback windows and forecast horizons as tasks, learning mappings from time indices to values that generalize effectively. (You et al., 2021) treats historical and future data as tasks and utilizes Model-Agnostic Meta-Learning (MAML) (Finn et al., 2017) to adapt the model's parameters. Techniques initially developed for sequential domain adaptation and generalization, such as DDG-DA (Li et al., 2022) and TDG (Liu et al., 2023a), are also applicable to online time series forecasting, where they adapt the model to changing data distributions in various ways. In addition to meta-learning approaches, self-adaptive methods (Arik et al., 2022) employ backcasting self-supervised objectives to enhance a model's adaptability to future distributions during prediction. FSNet (Pham et al., 2023) introduces per-layer adapters and associative memory to dynamically adjust layer outputs and facilitate the learning of evolving patterns. RevIN (Nie et al., 2023) is a simple yet effective model-agnostic approach, which normalizes input samples before modeling and then reverses the process after making predictions. While these techniques collectively offer a rich landscape of strategies to mitigate the challenges posed by concept drift, they fall short in adequately considering the specialties of concept drift in time series data.

## 3 METHODOLOGIES

### 3.1 PROBLEM DEFINITION: ONLINE TIME SERIES FORECASTING VIA META-LEARNING

**Time series forecasting.** A time series is an ordered sequence of observations denoted as $\mathcal{Z} = \{z_1, z_2, \cdots, z_T\} \in \mathbb{R}^{T \times c}$, where $T$ is the total number of time steps, $z_i \in \mathbb{R}^c$, and $c$ is dimensionality. Time series forecasting aims at learning a prediction model $f_{\boldsymbol{\theta}}$ to predict the next $O$-steps at time $t$ given a look-back window of length $I$ as $z_{t-I+1:t} \xrightarrow{f(\cdot;\boldsymbol{\theta})} z_{t+1:t+O}$, where $\boldsymbol{\theta}$ is the parameter of the prediction model. For simplicity, in the remainder of the paper, we denote the input $z_{t-I+1:t}$ and the output $z_{t+1:t+O}$ as $\mathbf{x}$ and $\mathbf{y}$ and a set of input and output as $\mathbf{X}$ and $\mathbf{Y}$, respectively.

**Online time series forecasting.** In real-world applications, time series data often arrives in a streaming fashion with frequent concept drift. Traditional approaches training a prediction model once and evaluating it separately are unsuitable. Instead, a more appropriate approach is to continuously learn or update the model over a sequence of periods. At each period $p$, the model accesses and learns from the most recent data available. It then uses this knowledge to make predictions for future periods. Subsequently, the ground truth is revealed, allowing for updating the model for prediction in the upcoming period. In online forecasting settings, the model is iteratively updated at regular

Figure 2: Illustration of online time series forecasting via LEAF. LEAF is a meta-model that guides the optimization of the prediction model $f_{\boldsymbol{\theta}}$.

periods, referred as online forecasting period, and then evaluated on the next period of incoming data. This cyclic process ensures that the model remains up-to-date and adapts to the changing nature of the time series. Moreover, online forecasting often does not start from scratch. In most cases, we warm up the model based on historical data. Figure 2 provides a visual representation of the procedure involved in online time series forecasting.

**Meta-learning.** Note that simply fitting a prediction model to the most recent observed data is insufficient for online time series forecasting, as the model needs to effectively generalize to future data that may exhibit varying characteristics due to concept drift. Adapting the prediction model to future data requires a meta-learning approach. Meta-learning, also known as learning to learn, is a machine learning approach where a model learns how to learn from experience or previous tasks, enabling it to quickly adapt and generalize to new tasks. In the context of online time series forecasting, our objective is to learn how to adapt to future data. Our meta-learning based LEAF algorithm is trained over a sequence of online forecasting periods, referred as the meta-training phase, and evaluated over another sequence of periods called the meta-test phase. The procedure of meta-learning for online forecasting is depicted in Figure 2.

## 3.2 LEARNING TO EXTRAPOLATE AND ADJUST

**Model Overview.** We first introduce notations and outline the objective of LEAF. At each online forecasting period $p$, we have a training set $\mathcal{D}_{\text{train}}^p = \{\mathbf{X}_p, \mathbf{Y}_p\}$ and a test set $\mathcal{D}_{\text{test}}^p = \{\tilde{\mathbf{X}}_p, \tilde{\mathbf{Y}}_p\}$. Without loss of generality, we assume that both the training set and the test set contain an equal number of samples, with a total of $B$ samples. Specifically, $\mathbf{X}_p = \{\mathbf{x}_p^{(i)}\}_{i=1}^B$ and $\tilde{\mathbf{X}}_p = \{\tilde{\mathbf{x}}_p^{(i)}\}_{i=1}^B$. The main objective of LEAF is to leverage knowledge gained from historical periods and generate model parameters to make accurate forecasts on $\mathcal{D}_{\text{test}}^p$ in the presence of concept drift. To achieve this goal, LEAF learns two functions: extrapolation $\mathcal{E}(\cdot; \boldsymbol{\phi}_e)$ and adjustment $\mathcal{A}(\cdot; \boldsymbol{\phi}_a)$, addressing macro- and micro-drift, respectively, to generate parameters $\boldsymbol{\theta}_p$ of the prediction model at period $p$. The objective can be formulated as:

$$\min_{\boldsymbol{\phi}_e, \boldsymbol{\phi}_a} \sum_{i=1}^B \mathcal{L}(f(\tilde{\mathbf{x}}_p^{(i)}; \boldsymbol{\theta}_p^{(i)}); \tilde{\mathbf{y}}_p^{(i)}),$$
$$\text{s.t., } \boldsymbol{\theta}_p^{(i)} = \mathcal{A}(\boldsymbol{\theta}_p, \tilde{\mathbf{x}}_p^{(i)}; \boldsymbol{\phi}_a), \ \boldsymbol{\theta}_p = \mathcal{E}(\boldsymbol{\theta}_{p-k:p-1}^*; \boldsymbol{\phi}_e),$$

(1)

where the extrapolation function $\mathcal{E}(\cdot; \boldsymbol{\phi}_e)$ is used to anticipate the macro-drift which takes as the optimal parameters $\boldsymbol{\theta}_{p-k:p-1}^*$ from the previous $k$ periods as input and infer parameters $\boldsymbol{\theta}_p$ for period $p$, the adjustment function performs sample-specific parameter adjustment considering micro-drift within each sample, and $\mathcal{L}(\cdot)$ is the prediction loss. Figure 3(b) illustrates the framework of LEAF.

### 3.2.1 LEARNING TO EXTRAPOLATE IN LATENT SPACE

During the extrapolation stage, the meta-model LEAF aims at inferring a prediction model based on previously optimal models. It focuses on generating overall suitable model parameters for the next period by tracking and anticipating the macro-drift between different periods of data. However, we have observed that although LEAF is used to generate a bulk of model parameters, it does not need to operate on high-dimensional parameter spaces. This is because model parameters are used to depict the complex data patterns, most of which remain invariant, and only a small part will experience drift. In other words, the concept drift can be effectively modeled in a low-dimensional latent space. As a result, we introduce a latent embedding $\boldsymbol{H}$ which can be decoded to the model parameter $\boldsymbol{\theta}$, and the extrapolation is performed on this latent space.

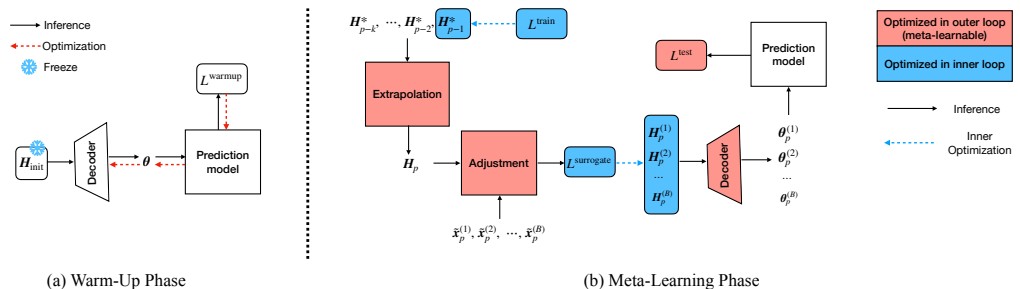

(a) Warm-Up Phase             (b) Meta-Learning Phase

Figure 3: Model Architecture. (a) Warm-up of the target prediction model. The initial latent embedding $\boldsymbol{H}_{\text{init}}$ is randomly generated and frozen during the warm-up phase, and the encoder is learned with prediction loss on warm-up dataset. (b) Meta-learning of LEAF which consists of three meta-learnable modules: extrapolation, adjustment, and parameter decoder.

At period $p$, we have $\mathcal{D}^p_{\text{train}} = \{\mathbf{X}_p, \mathbf{Y}_p\}$ and $\mathcal{D}^p_{\text{test}} = \{\tilde{\mathbf{X}}_p, \tilde{\mathbf{Y}}_p\}$. Notably, $\mathcal{D}^p_{\text{train}}$ is identical to the test data $\mathcal{D}^{p-1}_{\text{test}}$, as shown in Figure 2. In other words, if we optimize the prediction model on $\mathcal{D}^p_{\text{train}}$, we can obtain the optimal model at period $p-1$ in the latent space as follows:

$$\boldsymbol{H}^*_{p-1} = \text{SGD}(\boldsymbol{H}_{p-1}, \mathcal{L}^{\text{train}}, \mathcal{D}^p_{\text{train}}), \tag{2}$$

where SGD represents gradient descent on $\mathcal{D}^p_{\text{train}}$ with respect to prediction loss. This optimization process leads to the optimal latent embedding $\boldsymbol{H}^*_{p-1}$. Subsequently, we introduce an LSTM that infers $\boldsymbol{H}_p$ based on the previous $k$ optimal latent embeddings $\boldsymbol{H}^*_{p-k:p-1}$ and a decoder to generate model parameters, which can be formulated as:

$$\boldsymbol{H}_p = \text{LSTM}(\boldsymbol{H}^*_{p-k:p-1}), \quad \boldsymbol{\theta}_p = \text{Decoder}(\boldsymbol{H}_p). \tag{3}$$

Moving forward period $p+1$, $\mathcal{D}^p_{\text{test}}$ (or $\mathcal{D}^{p+1}_{\text{train}}$) is revealed. The above steps can be repeated to obtain $\boldsymbol{H}^*_p$, $\boldsymbol{H}_{p+1}$ and $\boldsymbol{\theta}_{p+1}$.

### 3.2.2 LEARNING TO ADJUST VIA SURROGATE LOSS

The inferred model in the extrapolation stage is designed to anticipate and account for macro-drift. However, they do not consider the micro-drift that can occur within each individual sample. To address this issue, we introduce a meta-learnable surrogate loss that adjusts or modulates the latent embedding $\boldsymbol{H}_p$ for each sample $\{\tilde{\mathbf{x}}^{(i)}_p\}^B_{i=1}$ in $\mathcal{D}^p_{\text{test}}$ to obtain sample-specific embeddings $\{\boldsymbol{H}^{(i)}_p\}^B_{i=1}$. This surrogate loss is implemented using neural networks and applied during the testing phase, enabling the adjustment of the latent embedding based on sample-specific characteristics.

The rationale behind this approach is that micro-drift is often caused by external events, such as weather conditions or holidays. Although data affected by micro-drift may exhibit different patterns, they tend to deviate from their context in a similar manner. For example, the electricity consumption pattern during the summer typically differs from that in the winter, while during holidays it tends to decrease regardless of the season. By incorporating a meta-learnable surrogate loss, we aim to capture and adjust for these sample-specific deviations caused by micro-drift. Specifically, during the meta-training phase, the surrogate loss is learned to capture and compensate for different types of deviations. During the meta-testing phase, the model utilizes the learned surrogate loss to adjust the latent embedding based on the specific characteristics of each individual sample. To achieve this, the loss network takes into account the following three factors to capture sample-specific micro-drift:

**Sample and base prediction.** The sample itself $\tilde{\mathbf{x}}^{(i)}_p$ and the base prediction with inferred model parameters in extrapolation stage $f(\tilde{\mathbf{x}}^{(i)}_p; \boldsymbol{\theta}_p)$ are introduced as basic characteristics.

**Base latent embedding.** Another important part of information is the prediction model. As we optimize the model in a latent space, the inferred latent embedding in the extrapolation stage $\boldsymbol{H}_p$ is passed to the loss network, providing the contextual information and underlying pattern.

**Sample's relationship to training data.** Furthermore, we introduce a relation network to account for sample's relationship to its context. The relation network $\mathcal{R}(\cdot)$ receives embeddings of the sample $\tilde{\mathbf{x}}^{(i)}_p$ and training set $\mathcal{D}^p_{\text{train}}$ and returns a vector representing their relationship:

$$\boldsymbol{R}^{(i)}_p = \mathcal{R}\left(g(\tilde{\mathbf{x}}^{(i)}_p), \mathbb{E}_{\mathbf{x}^{(j)}_p \sim \mathcal{D}^p_{\text{train}}} g(\mathbf{x}^{(j)}_p)\right), \tag{4}$$

---

**Algorithm 1:** Training Procedure of LEAF

---

**Input:** Number of meta-training periods $P$, data $\{\mathcal{D}_{\text{train}}^p, \mathcal{D}_{\text{test}}^p\}_{p=1}^P$, learning rates $\eta, \alpha, \mu$, number of historical periods in extrapolation stage $k$, prediction model $f(\cdot)$

1  Randomly initialize the parameters $\phi_e, \phi_a$, and $\omega$ of LEAF ;
2  Let $\psi = \{\phi_e, \phi_a, \omega\}$ ;
3  Randomly initialize the latent embedding $\boldsymbol{H}_0$ ;
4  Create a queue $\mathcal{Q}$ of size $k$ storing the recent $k$ optimal latent embeddings;
5  Initialize $\mathcal{Q}$ as $\mathcal{Q} = [\boldsymbol{H}_0, \cdots, \boldsymbol{H}_0]$ ;
6  **for** $p = 1, 2, \cdots, P$ **do**
7    Evaluate inner-loop prediction loss $\mathcal{L}_p^{\text{train}}$ on training dataset $\mathcal{D}_{\text{train}}^p$ ;
8    Perform gradient descent $\boldsymbol{H}_{p-1}^* = \boldsymbol{H}_{p-1} - \eta \nabla_{\boldsymbol{H}_{p-1}} \mathcal{L}_p^{\text{train}}$;
9    $\mathcal{Q}.\text{Deque}().\text{Enque}(\boldsymbol{H}_{p-1}^*.\text{detach}())$
10   Compute $\boldsymbol{H}_p$ and $\boldsymbol{\theta}_p$ using Equation (3);   /* Extrapolation stage */
11   **for** $\tilde{\mathbf{x}}_p^{(i)}$ *in* $\tilde{\mathbf{X}}_p$ **do**
12     /* Adjustment stage: traverse all test inputs, and conduct sample-specific adjustment */
13     Compute $\mathcal{L}^{\text{surrogate}}(\tilde{\mathbf{x}}_p^{(i)})$ using Equation (5);
14     $\boldsymbol{H}_p^{(i)} = \boldsymbol{H}_p - \alpha \nabla_{\boldsymbol{H}_p} \mathcal{L}^{\text{surrogate}}(\tilde{\mathbf{x}}_p^{(i)})$;
15     $\boldsymbol{\theta}_p^{(i)} = \text{Decoder}(\boldsymbol{H}_p^{(i)})$, and load $\boldsymbol{\theta}_p^{(i)}$ into $f(\cdot)$;
16     Evaluate prediction loss $\mathcal{L}_{p,i}^{\text{test}}$ w.r.t. $\tilde{\mathbf{x}}_p^{(i)}$ ;
17   **end**
18   Compute $\mathcal{L}^{\text{LEAF}}$ using Equation (7);
19   Perform gradient descent $\psi = \psi - \mu \nabla_\psi \mathcal{L}^{\text{LEAF}}$;
20 **end**

---

where $g(\cdot)$ is an embedding function, and the training data embedding is computed as the mean pooling of embeddings of all samples in training set. The relation network captures the similarities or dissimilarities in their patterns, which is valuable in capturing the sample-specific micro-drift.

To sum up, the final loss network $s(\cdot)$ are defined as

$$\mathcal{L}^{\text{surrogate}}(\tilde{\mathbf{x}}_p^{(i)}) = s\left(\tilde{\mathbf{x}}_p^{(i)}, f(\tilde{\mathbf{x}}_p^{(i)}; \boldsymbol{\theta}_p), \boldsymbol{H}_p, \boldsymbol{R}_p^{(i)}\right). \tag{5}$$

The surrogate loss guides the adjustment of the latent embedding for each sample using gradient descent, which can be further decoded to obtain sample-specific model parameters:

$$\boldsymbol{H}_p^{(i)} = \boldsymbol{H}_p - \alpha \nabla_{\boldsymbol{H}_p} \mathcal{L}^{\text{surrogate}}(\tilde{\mathbf{x}}_p^{(i)}), \quad \boldsymbol{\theta}_p^{(i)} = \text{Decoder}(\boldsymbol{H}_p^{(i)}), \tag{6}$$

where $\alpha$ is the learning rate, and $\boldsymbol{H}_p^{(i)}$ and $\boldsymbol{\theta}_p^{(i)}$ stand for sample-specific latent embedding and model parameters, respectively. Finally, the sample-specific parameters are loaded into the model to make forecasts using $f(\tilde{\mathbf{x}}_p^{(i)}); \boldsymbol{\theta}_p^{(i)})$.

### 3.2.3 MODELING LEARNING

In this subsection, we outline the training procedure of LEAF, as depicted in Algorithm 1. We denote the parameters of extrapolation module and adjustment module as $\phi_e$ and $\phi_a$, respectively. Additionally, $\omega$ represents the parameter of the decoder.

At each period $p$, as shown in Figure 3(b), after extrapolation and adjustment stages (the "inner loop"), we make predictions on $\mathcal{D}_{\text{test}}^p$. The test set is then utilized to update the parameters of meta-learners $\phi_a, \phi_e, \omega$ (the "outer loop"). More precisely, the optimization process of "outer loop" is performed by minimizing the following objective:

$$\mathcal{L}^{\text{LEAF}} = \min_{\phi_e, \phi_a, \omega} \sum_{i=1}^B \mathcal{L}^{\text{test}}(f(\tilde{\mathbf{x}}_p^{(i)}; \boldsymbol{\theta}_p^{(i)}); \tilde{\mathbf{y}}_p^{(i)}) + \gamma \|\text{stopgrad}(\boldsymbol{H}_p^*) - \boldsymbol{H}_p\|_2^2, \tag{7}$$

where the first term is the prediction loss on the test set, the second term regularizes the extrapolation in Equation (3) to output latent embedding that is close to the optimal latent embedding, and $\gamma$ is the coefficient of the regularization.

Recall that online forecasting often involves a warm-up phase. Since we optimize the parameters of the prediction model in the latent space, the traditional training strategy is not directly applicable. To address this issue, we randomly generate a latent embedding $\boldsymbol{H}_{\text{init}}$, which is then decoded into parameters of prediction model using the decoder. During the training procedure of the warm-up

phase, $H_{\text{init}}$ remains fixed and the decoder is trained using the prediction loss on warm-up dataset. Subsequently, at the onset of the meta-learning, we initialize $H_0$ with $H_{\text{init}}$ and $\omega$ with the learned decoder parameter from warm-up phase. Figure 3(a) illustrates the warm-up strategy.

**Remarks.** LEAF generates the parameters for the last few layers of the target prediction model. It is important to note that LEAF is a model-agnostic framework, allowing for the application of different types of layers. For instance, when using DLinear as the target model, we generate the parameters for a linear layer. In the case of PatchTST, we generate the parameters for the last transformer block, which includes the Query/Key/Value projection networks and a feed-forward network. To apply LEAF to different types of layers, we require prior knowledge of the parameter shapes of the network layers. By calculating the total number of parameters that need to be generated, we can determine the width of the decoder in LEAF. The generated parameters are then appropriately reshaped and loaded into the respective layers of the target prediction model. By employing this approach, LEAF is able to adapt to various types of layers in the target prediction model, allowing for flexibility and compatibility across different network architectures.

## 4 EXPERIMENTS

In this section, we conduct experiments to investigate the performance of LEAF compared with existing algorithms, focusing on the following research questions: (**RQ1**) Can LEAF outperform SOTA model-agnostic concept drift adaptation methods in online time series forecasting scenarios? (**RQ2**) How do different components of LEAF contribute to resolving concept drift problems?

### 4.1 EXPERIMENTAL SETTINGS

**Datasets.** We evaluate our method on six time series forecasting datasets. (1) **ETT-small** [2] (Zhou et al., 2021) dataset contains observations of oil temperature along with six power load features over two years. We take into consideration the ETTm1 benchmark, wherein the observations are recorded on an hourly basis, as well as the ETTh2 benchmark, which records observations at 15-minute intervals. For ETTh2, the model update interval (number of timesteps in each $\mathcal{D}_{\text{train}}$) is 288, the look-back window is 96 and the forecast horizon is 24. For ETTm1, the model update interval is 672, the look-back window is 288 and the forecast horizon is 24. (2) **ECL**[3] (Electricity Consuming Load) dataset records hourly electricity consumption of 321 users over three years. We random sample 12 users. The model update interval is 288, the look-back window is 96 and the forecast horizon is 24. (3) **Load** dataset contains three real-world univariate electric load benchmarks in different types of areas at 15-minute intervals from 2020 to 2022. Figure 5 and Figure 6 ( in Appendix A.1) visually demonstrate the presence of concept drift within each benchmark, including diverse types of macro- and micro-drifts. These drifts arise due to multifaceted factors such as intricate weather conditions (e.g., extreme weather events), social dynamics (e.g., adjustments in factory production plans, increased utilization of renewable energy sources, population growth, and holidays), as well as political influences (e.g., the impact of COVID-19 and changes in electricity price). These factors, characterized by their complexity, cannot be comprehensively captured or precisely quantified numerically. Consequently, forecasters must endeavor to model the evolving hidden dynamics solely based on the available load data.

**Baselines.** We compare our method with four model-agnostic baselines, including: (1) **Naïve** that only trains model on warm-up dataset and freeze henceforth, (2) **Naïve** † that trains on warm-up and meta-train dataset and freeze at meta-test, (3) **Retrain** that updates the last layer of backbone model at each period by gradient descent using all available data, (4) **Fine-tune** that updates the last layer of backbone model at each period by gradient descent using only the data in the period, (5) **ER** (Chaudhry et al., 2019) that employs a memory bank of most recent samples, and (6) **DER++** (Buzzega et al., 2020) that is a variate of ER with knowledge distillation. We also compare our method with **FSNet** (Pham et al., 2023), a recent SOTA for concept drift in online time series forecasting that uses TCN (Bai et al., 2018) as the backbone and incorporates an experience replay. Since few model-agnostic methods exist for alleviating concept drift in online time series forecasting, we use ER and DER++ from continual learning. We show empirically that ER and DER++ are competitive baselines in online time series forecasting scenarios (Pham et al., 2023).

---

[2] https://github.com/zhouhaoyi/ETDataset
[3] https://archive.ics.uci.edu/dataset/321/electricityloaddiagrams20112014

Table 1: The final comparison performance averaged over meta-test and five random seeds. The bold values are the best results.

| Model | Method | Load-1 | | Load-2 | | Load-3 | | ETTh2 | | ETTm1 | | ECL | |
|---|---|---|---|---|---|---|---|---|---|---|---|---|---|
| | | MSE | MAE | MSE | MAE | MSE | MAE | MSE | MAE | MSE | MAE | MSE | MAE |
| LSTM | Naïve | 2.0479 | 1.0194 | 2.0147 | 1.0076 | 3.9826 | 1.4059 | 0.8073 | 0.6296 | 2.2222 | 0.9538 | 0.2406 | 0.3399 |
| | Naïve† | 1.4887 | 0.8487 | 1.6944 | 0.9192 | 2.9772 | 1.1576 | 0.9171 | 0.6414 | 1.4304 | 0.7529 | 0.1794 | 0.2943 |
| | Retrain | 1.5145 | 0.8507 | 1.6379 | 0.8879 | 2.9498 | 1.1489 | 0.9011 | 0.6396 | 1.4140 | 0.7331 | 0.1723 | 0.2884 |
| | Fine-tune | 1.4838 | 0.8545 | 1.5356 | 0.8553 | 3.0218 | 1.1342 | 0.7729 | 0.6114 | 1.4362 | 0.750 | 0.1629 | 0.2700 |
| | ER | 1.4722 | 0.8570 | 1.4737 | 0.8346 | 2.9411 | 1.1192 | 0.8920 | 0.6359 | 1.4351 | 0.7409 | 0.1660 | 0.2828 |
| | DER++ | 1.4839 | 0.8622 | 1.4858 | 0.8393 | 2.9266 | 1.1142 | 0.8841 | 0.6338 | 1.4390 | 0.7423 | 0.1674 | 0.2839 |
| | LEAF | **0.6150** | **0.4765** | **0.6426** | **0.5064** | **1.7515** | **0.8084** | **0.7398** | **0.5892** | **0.7246** | **0.5397** | **0.1216** | **0.2328** |
| DLinear | Naïve | 0.6739 | 0.5056 | 0.6445 | 0.4827 | 1.9530 | 0.8812 | 0.6818 | 0.5609 | 0.7419 | 0.5388 | 0.1447 | 0.2364 |
| | Naïve† | 0.6371 | 0.4813 | 0.6233 | 0.4739 | 1.8153 | 0.8165 | 0.7429 | 0.5721 | 0.6666 | 0.5006 | 0.1365 | 0.2284 |
| | Retrain | 0.6329 | 0.4803 | 0.6214 | 0.4763 | 1.7959 | 0.8103 | 0.7498 | 0.5721 | 0.6668 | 0.4987 | 0.1363 | 0.2275 |
| | Fine-tune | 0.7172 | 0.5217 | 0.6738 | 0.5068 | 2.1112 | 0.9038 | 0.6465 | 0.5493 | 0.7173 | 0.5200 | 0.1380 | 0.2291 |
| | ER | 0.6246 | 0.4728 | 0.6158 | 0.4802 | 1.7930 | 0.8098 | 0.7511 | 0.5733 | 0.6654 | 0.4978 | 0.1359 | 0.2275 |
| | DER++ | 0.6241 | 0.4723 | 0.6151 | 0.4786 | 1.7921 | 0.8091 | 0.7480 | 0.5725 | 0.6642 | 0.4976 | 0.1358 | 0.2273 |
| | LEAF | **0.6042** | **0.4605** | **0.5915** | **0.4590** | **1.6952** | **0.7742** | **0.6457** | **0.5491** | **0.6161** | **0.4836** | **0.1126** | **0.2200** |
| PatchTST | Naïve | 2.5162 | 1.1297 | 0.9509 | 0.6813 | 2.4200 | 1.0442 | 0.8120 | 0.6244 | 1.4665 | 0.7599 | 0.1516 | 0.2501 |
| | Naïve† | 0.7630 | 0.5818 | 0.6674 | 0.5216 | 1.8485 | 0.8729 | 0.7378 | 0.5869 | 0.6579 | 0.5136 | 0.1120 | 0.2214 |
| | Retrain | 0.6678 | 0.5318 | 0.6471 | 0.5206 | 1.7869 | 0.8608 | 0.7260 | 0.5861 | 0.6395 | 0.5036 | 0.1112 | 0.2193 |
| | Fine-tune | 0.9482 | 0.6445 | 0.8756 | 0.6265 | 2.7394 | 1.1105 | 0.6919 | 0.5740 | 0.8853 | 0.5944 | 0.1159 | 0.2244 |
| | ER | 0.6771 | 0.5370 | 0.6353 | 0.5131 | 1.7507 | 0.8348 | 0.7232 | 0.5823 | 0.6479 | 0.5067 | 0.1106 | 0.2190 |
| | DER++ | 0.6705 | 0.5332 | 0.6338 | 0.5108 | **1.7388** | **0.8300** | 0.7213 | 0.5808 | 0.6442 | 0.5052 | 0.1104 | 0.2187 |
| | LEAF | **0.6429** | **0.5165** | **0.6155** | **0.5054** | 1.9582 | 0.8794 | **0.6707** | **0.5640** | 0.6717 | 0.5082 | **0.1098** | **0.2161** |
| TCN | Naïve | 1.3692 | 0.8554 | 0.8599 | 0.6093 | 2.3085 | 1.0086 | 0.9151 | 0.6854 | 1.9414 | 0.9073 | 0.2916 | 0.3718 |
| | Naïve† | 1.1528 | 0.7598 | 0.7877 | 0.5770 | 1.9786 | 0.8871 | 0.9194 | 0.6676 | 1.1733 | 0.7067 | 0.2138 | 0.3230 |
| | Retrain | 1.0901 | 0.7308 | 0.7956 | 0.2759 | 2.0241 | 0.8921 | 0.9212 | 0.6676 | 1.2118 | 0.7132 | 0.1912 | 0.3076 |
| | Fine-tune | 1.2864 | 0.7807 | 0.9606 | 0.6523 | 2.3098 | 0.9889 | 0.7911 | 0.6271 | 1.3077 | 0.7544 | 0.1887 | 0.2911 |
| | ER | 1.1033 | 0.7447 | 0.8044 | 0.6089 | 2.1006 | 0.9430 | 0.9434 | 0.6684 | 1.2573 | 0.7216 | 0.1859 | 0.3023 |
| | DER++ | 1.1110 | 0.7495 | 0.8108 | 0.6120 | 2.0919 | 0.9387 | 0.9112 | 0.6585 | 1.1896 | 0.6995 | 0.2203 | 0.3267 |
| | FSNet | 0.8024 | 0.5657 | 0.7221 | 0.5488 | 2.2942 | 0.9489 | 1.4468 | 0.8292 | 0.9761 | 0.6352 | 0.1658 | 0.2840 |
| | LEAF | **0.7080** | **0.5312** | **0.6934** | **0.5299** | **1.8872** | **0.8858** | **0.7214** | **0.5887** | **0.7727** | **0.5526** | **0.1340** | **0.2499** |

**Implementation Details.** For all benchmarks, we split the data into warm-up/meta-train/meta-test by the ratio of 0.1/0.6/0.3. In warm-up phase, we use Adam (Kingma & Ba, 2014) with fixed learning rate of 0.001 to optimize the prediction model w.r.t $l_2$ (Mean Squared Loss) loss. The warm-up epoch is 10 and warm-up batch size is 32. In meta-training and meta-testing phases, at each period $p$, we use Adam with learning rate of 0.001 to obtain the optimal latent embedding and update parameters of meta-learners, the update epoch is 50 and 1 respectively. We implement Decoder, relation network $\mathcal{R}(\cdot)$, and loss network $s(\cdot)$ with MLPs. We perform cross-validation on meta-train to select appropriate hyper-parameters for all methods. We notice that LEAF is able to achieve competitive results with minor hyper-parameter tuning. We report the hyperparameter sensitivity in Appendix A.2. Besides, all baselines are well-tuned using Bayesian optimization algorithm in Neural Network Intelligence toolkit[4]. All experimental results are the average of the five independent trials with different random seeds.

## 4.2 PERFORMANCE COMPARISON (RQ1)

The results of comparison performance are shown in Table 1. We apply model-agnostic methods to four types of prediction models, including LSTM (Hochreiter & Schmidhuber, 1997), DLinear (Zeng et al., 2023), PatchTST (Nie et al., 2023) and TCN (Bai et al., 2018). We place FSNet (Pham et al., 2023) in TCN category since it employs a TCN-based backbone. We use mean squared error (MSE) and mean absolute error (MAE) as evaluation metrics and they are averaged over meta-test periods. From Table 1, we can observe that: (1) our proposed LEAF significantly improves forecast performance in almost all cases, especially in Load benchmarks, for different types of prediction models. Overall, LEAF has a 17.8% reduction in error compared with second-best method. Especially, LEAF has an average decrease of $45.2\%$ in MSE when using LSTM as prediction model and $22.3\%$ when using TCN; (2) LEAF performs surprisingly well on Load-1, Load-2, and ETTm1 datasets, while the advantage is not so obvious on ECL dataset. This is because time series from Load-1, Load-2, and ETTm1 datasets exhibit much stronger and more complex concept drift; (3) ER and DER++ show competitive performance in ECL dataset, as time series in this dataset holds strong seasonality and recurring patterns. These methods incorporate a memory of samples which helps alleviate catastrophic forgetting and remembering recurring patterns in history. We note here ER and its variant are orthogonal to our work which can be embedded easily into LEAF.

---

[4]https://nni.readthedocs.io/en/stable

Table 2: Ablation study results on Load-1 and ETTm1 with prediction models of DLinear and PatchTST. The bold values are the best results.

| Model | Method | Load-1 | | ETTm1 | |
|---|---|---|---|---|---|
| | | MSE | MAE | MSE | MAE |
| DLinear | Fine-tune | 0.7172 | 0.5217 | 0.7173 | 0.5200 |
| | Latent fine-tune | 0.6683 | 0.5057 | 0.7019 | 0.5196 |
| | Latent fine-tune + A | 0.6562 | 0.5045 | 0.6745 | 0.5295 |
| | EN | 0.6578 | 0.4931 | 0.6723 | 0.5066 |
| | EN + A | **0.6042** | **0.4605** | **0.6161** | **0.4836** |
| PatchTST | Fine-tune | 0.9482 | 0.6445 | 1.4665 | 0.7599 |
| | Latent fine-tune | 0.8103 | 0.6068 | 0.7546 | 0.5473 |
| | Latent fine-tune + A | 0.6783 | 0.5198 | 0.6823 | 0.5143 |
| | EN | 0.7539 | 0.5800 | 0.7334 | 0.5410 |
| | EN + A | **0.6429** | **0.5165** | **0.6717** | **0.5082** |

## 4.3 ABLATION STUDIES (RQ2)

We conduct ablation studies of LEAF to validate the effectiveness of extrapolation and adjustment stages. We evaluate the performance of four variants of LEAF, utilizing DLinear and PatchTST as target prediction model on Load-1 and ETTm1 datasets. Considering that LEAF can be seen as an advanced method of fine-tuning, we begin with the most basic fine-tune method and gradually incorporate designed modules in LEAF to construct the model variants. The variants includes: (1) **Fine-tune** that is foundation of LEAF where the optimization is performed in the parameter space, (2) **Latent fine-tune** that uses an average of last five $H^*$ instead of the extrapolated latent embedding $H$, (3) **EN** that introduces the extrapolation module as described in Section 3.2.1 with $k = 5$, (4) **Latent fine-tune + A** that incorporates the adjustment stage on **Latent fine-tune**, and (5) **EN + A** that incorporates the adjustment stage on extrapolation module and is identical to standard LEAF.

The results of ablation studies are shown in Table 2. We observe first that fine-tuning model in the latent space (**Latent fine-tune**) can significantly improve the forecast performance in almost all benchmarks. This outcome verifies that optimizing model in a low-dimensional latent space is rational. Furthermore, **EN** introducing our proposed extrapolation module surpasses the performance of **Latent fine-tune**, thereby confirming its effectiveness in extrapolating the macro-drift. Lastly, the inclusion of the sample-specific adjustment yields a further enhancement in predictive performance, demonstrating the effectiveness of this stage in alleviating micro-drift. Moreover, we plot prediction results of different variants in Figure 7 (see Appendix A.3)

## 4.4 CASE STUDIES

In this section, we examine the forecast ability of LEAF under macro-drift and micro-drift by case studies. Figure 4 illustrates the predictions of different methods on four slices of Load-1 using LSTM as predction model. The first two rows demonstrate cases wherein there exists a drift in the pattern of load within a period. Specifically, two load usage peaks are gradually transition into a singular peak which may be caused by alterations in weather conditions. In both scenarios, LEAF successfully captures the micro-drift and generates precise forecasts, whereas other methods fail to adapt to such micro-drift. The third row illustrates a case involving long-term macro-drift characterized by a gradual modification in the trend and seasonality of the time series. In this case, LEAF can capture the gradually increasing trend as well as the change in seasonality. Conversely, even with the integration of RevIN (Nie et al., 2023), wherein the statistical attributes of the look-back window are incorporated into the forecasting process, the baseline methods continue to falter in adapting to the macro-drift. The final case exemplifies a combination of macro-drift and micro-drift, and in this instance, LEAF demonstrates considerably superior performance compared to the baseline methods.

In Appendix A.4, Figure 8 illustrates the trajectories of latent embedding $H$ and optimal latent embedding $H^*$ over the last 20 periods of Load-1. Notably, at each period, the predicted latent embedding $H$ (inferred during the extrapolation stage) consistently aligns closely with the optimal latent embedding, which serves as evidence of the extrapolating capabilities of our method in effectively tracking the evolving dynamics of data distributions.

## 5 CONCLUSION AND FUTURE WORK

In conclusion, our proposed LEAF framework offers a promising solution to the concept drift problem in online time series forecasting, specifically addressing both macro-drift and micro-drift. By integrating meta-learning, LEAF enhances the capabilities of deep prediction models by acquiring the ability to extrapolate and adapt to macro- and micro-drift, respectively. This model-agnostic

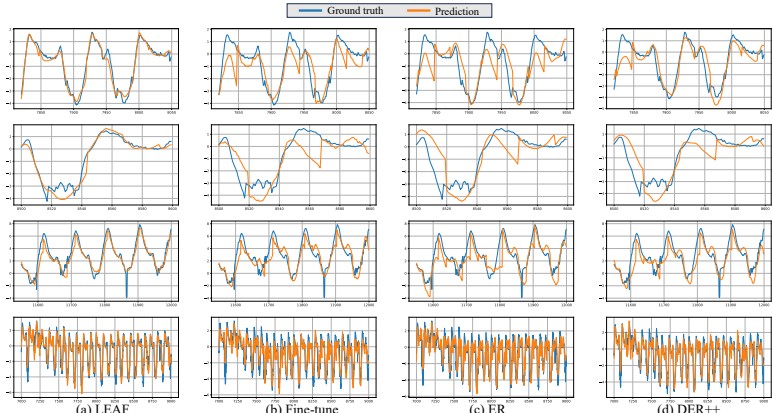

Figure 4: The predictions of LEAF, Fine-tune, ER, and DER++ using LSTM as prediction model on four slices of Load-1 dataset. The blue (orange) represents the ground truth (prediction).

framework can be applied to various deep prediction models, making it versatile and applicable in different domains. Extensive experiments on real-world benchmark datasets demonstrate the effectiveness of LEAF. The consistent superior performance of LEAF across different models and datasets highlights its ability to handle concept drift and improve forecasting accuracy in dynamic and evolving environments. In addition, we collected and open sourced three benchmark datasets with diverse and typical concept drifts. In the future, we plan to extend our framework to handle more intricate concept drift scenarios, and one potential direction is the combination of LEAF and continual learning methods.

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

# A APPENDIX

## A.1 INTRODUCTION OF LOAD DATASETS

**Load** dataset contains three real-world univariate electricity load benchmarks in different types of areas in China at 15-minute intervals from 2020 to 2022. Figure 5 and Figure 6 visually demonstrate the presence of concept drift within each benchmark, including diverse types of macro- and micro-drifts. These drifts arise due to multifaceted factors such as intricate weather conditions (e.g., extreme weather events), social dynamics (e.g., adjustments in factory production plans, increased utilization of renewable energy sources, population growth, and holidays), as well as political influences (e.g., the impact of COVID-19 and changes in electricity price). These factors, characterized by their complexity, cannot be comprehensively captured or precisely quantified numerically. Consequently, forecasters must endeavor to model the evolving hidden dynamics solely based on the available load data. The model update interval is 672, the look-back window is 288 and the forecast horizon is 24 for all benchmarks.

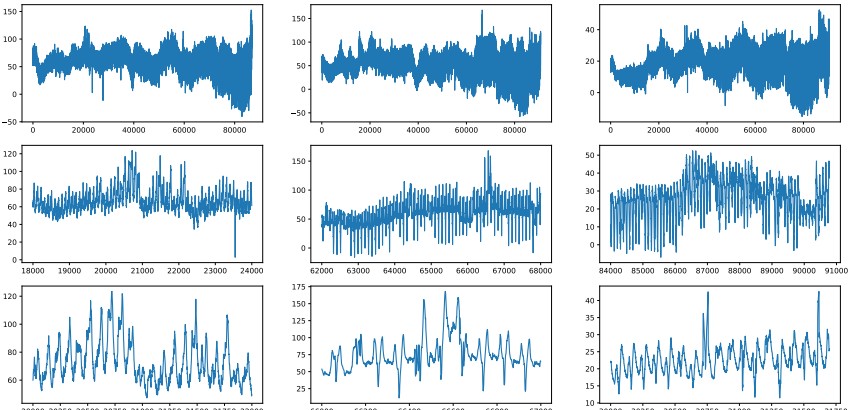

Figure 5: The Visualizations of Typical Concept Drift in Load Dataset. The observed concept drift within the dataset can be attributed to a confluence of intricate weather patterns, socio-economic factors, and political dynamics. While these factors are predictable in nature, their numerical representation as features poses significant challenges.

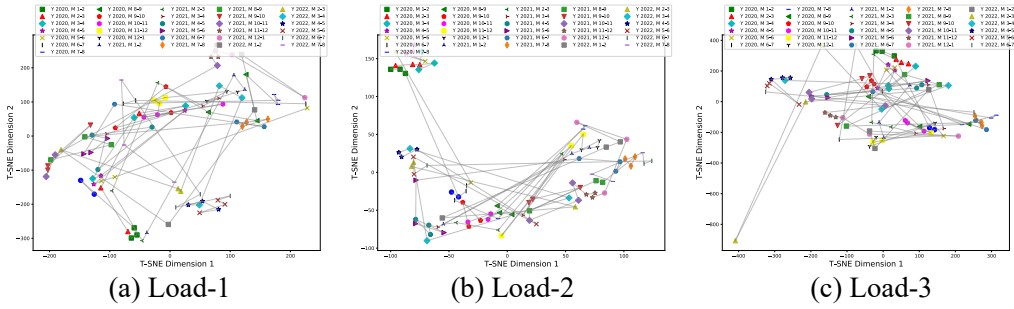

|  (a) Load-1 | (b) Load-2 | (c) Load-3 |

Figure 6: The visualizations of Three Load Datasets with t-SNE. All three Load datasets involve strong concept drift.

## A.2 STUDIES OF HYPERPARAMETER SENSITIVITY

Our proposed LEAF involves a few hyper-parameters, and we have not carefully tuned these parameters in our experiments. LEAF can consistently perform well under different hyper-parameter settings. Here, we run sensitivity experiments of three hyper-parameters, including the dimension

of $\boldsymbol{H}$, number of historical periods considered in extrapolation module $k$, and the coefficient of regularization term $\gamma$, on Load-1 and ETTm1 datasets for examples and report the numerical results in Table 3.It can be seen that LEAF can maintain its performance with a wide range of parameter values.

Table 3: Sensitivity analysis on hyper-parameters latent-dim (the dimension of latent embedding $\boldsymbol{H}$), $k$ and $\gamma$. The analysis is performed using DLinear as prediction model on Load-1 and ETTm1 benchmarks.

| Configuration | | | Load-1 | | ETTm1 | |
|---|---|---|---|---|---|---|
| latent-dim | $k$ | $\gamma$ | MSE | MAE | MSE | MAE |
| 32 | 3 | 0.1 | 0.5920 | 0.4551 | 0.6106 | 0.4807 |
| 32 | 10 | 0.1 | 0.5867 | 0.4550 | 0.6113 | 0.4809 |
| 32 | 3 | 0.2 | 0.5861 | 0.4521 | 0.6149 | 0.4820 |
| 32 | 10 | 0.2 | 0.5856 | 0.4552 | 0.6107 | 0.4815 |
| 64 | 3 | 0.1 | 0.6059 | 0.4664 | 0.6110 | 0.4817 |
| 64 | 10 | 0.1 | 0.6034 | 0.4608 | 0.6175 | 0.4830 |
| 64 | 3 | 0.2 | 0.5994 | 0.4582 | 0.6134 | 0.4828 |
| 64 | 10 | 0.2 | 0.6031 | 0.4659 | 0.6164 | 0.4833 |
| 128 | 3 | 0.1 | 0.6232 | 0.4666 | 0.6108 | 0.4813 |
| 128 | 10 | 0.1 | 0.6031 | 0.4616 | 0.6125 | 0.4828 |
| 128 | 3 | 0.2 | 0.6104 | 0.4668 | 0.6062 | 0.4800 |
| 128 | 10 | 0.2 | 0.6010 | 0.4606 | 0.6136 | 0.4830 |

## A.3 PLOTS OF ABLATION STUDIES

In addition to numeric results of ablation studies, we plot prediction results of different variants in Figure 7. We find that Fine-tune fail to capture details of data characteristics. By adding modules of latent optimization, extrapolation, and adjustment one by one, the algorithm gradually gains the ability to capture data details and concept drift.

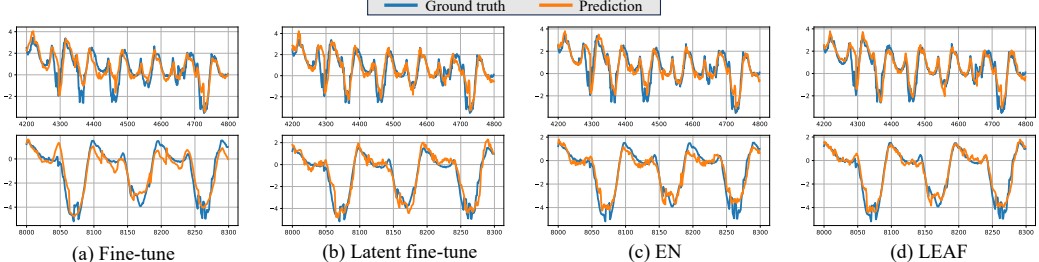

(a) Fine-tune    (b) Latent fine-tune    (c) EN    (d) LEAF

Figure 7: The predictions of different variants of LEAF using PatchTST as prediction model on Load-1 benchmark. The blue (orange) represents the ground truth (prediction).

## A.4 SUPPLEMENTARY CASE STUDIES

Figure 8 illustrates the trajectories of latent embedding $\boldsymbol{H}$ and optimal latent embedding $\boldsymbol{H}^*$ over the last 20 periods of Load-1. Notably, at each period, the predicted latent embedding $\boldsymbol{H}$ (determined during the extrapolation stage) consistently aligns closely with the optimal latent embedding, which serves as evidence of the extrapolating capabilities of our method in effectively tracking the evolving dynamics of data distributions.

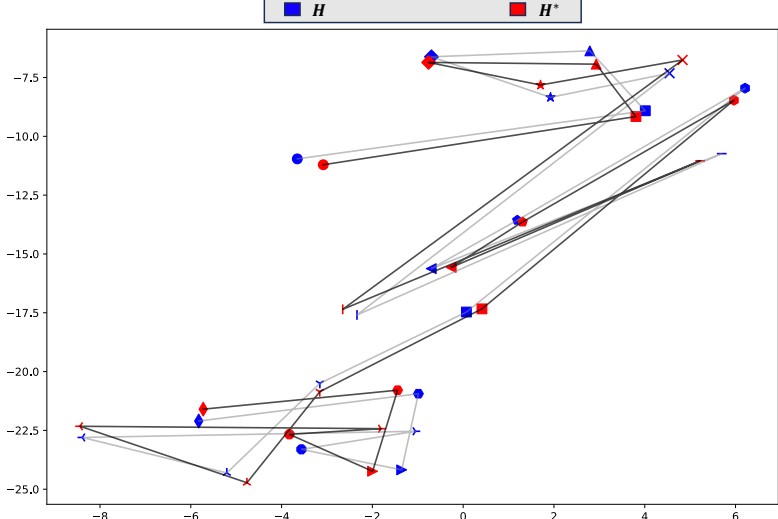

Figure 8: The Visualization of Latent Embeddings with t-SNE. The inferred latent embedding $H$ and optimal latent embedding $H^*$ over the last 20 periods of Load-1 with DLinear as prediction model. The blue (red) represents $H$ ($H^*$). Different periods are represented by different shapes and connected by black ($H^*$) and gray lines ($H$).

Table 4: The final comparison performance averaged over meta-test and five random seeds with forecast horizon 192. The bold values are the best results.

| Model | Method | Load-1 | | ETTm1 | |
|---|---|---|---|---|---|
| | | MSE | MAE | MSE | MAE |
| DLinear | Fine-tune | 1.0913 | 0.6439 | 1.0553 | 0.6503 |
| | ER | 1.0868 | 0.6416 | 1.0435 | 0.6462 |
| | DER++ | 1.0825 | 0.6392 | 1.0362 | 0.6435 |
| | LEAF | **1.0035** | **0.6210** | **1.0278** | **0.6470** |
| PatchTST | Fine-tune | 1.3368 | 0.7903 | 1.6296 | 0.8210 |
| | ER | 1.2561 | 0.7723 | 1.4456 | 0.7840 |
| | DER++ | 1.2386 | 0.7633 | 1.3812 | 0.7674 |
| | LEAF | **1.0967** | **0.6917** | **1.0570** | **0.6630** |

