# OpenReview forum: "Learning to Extrapolate and Adjust: Two-Stage Meta-Learning for Concept Drift in Online Time Series Forecasting"
_ICLR.cc/2024/Conference — Submitted to ICLR 2024_

### Official Review · Reviewer_RMU1 · 2023-10-21

**Soundness:** 2 fair
**Presentation:** 2 fair
**Contribution:** 2 fair
**Rating:** 5
**Confidence:** 4

**Summary:**

The paper proposed an online forecasting framework. It has two components to capture the macro and micro changes in the time series. An extrapolation network to predict the embedding of a time period based on optimal embedding from previous time periods. Then, the embedding is adjusted individually for each sample before using a decoder to generate the model parameters. The authors tested their methods on 6 datasets and compared them to several relevant continue learning baselines. An ablation study was done to show their impact on the performances. The authors also open sourced 3 datasets.

Disclaimer: I am not familiar with the continuing learning for forecasting literatures, and I would need to count on others to comment on the related works.

==== After reading the authors revision and response ===
I raised my score to 5 but I don't think the paper is good enough to be accepted at ICLR.

**Strengths:**

The paper tries to tackle a practical problem where forecasting models need to be updated given new data, continuously. It’s a bold attempt to combine many ideas: using embedding; decoding model parameters directly from embedding; using LSTM to predict embedding based on previous ones; adjust embeddings on a per-sample basis.

**Weaknesses:**

I could grasp the main ideas of the paper but some parts are not clear to me. For example, how exactly L_{surrogate} is computed? There are more in the question section.

Comparing Table 1 and Table 2, it seems the key to beat the baselines is the adjustment. Without it, the proposed method (which is quite complex) will not champion the others. This is not too surprising as the model is adjusted per-sample. But it comes with more computational cost and there is no address on this aspect in the paper.

Following from the above, the paper used many ideas and I think many of them, if studied thoroughly, can be a paper itself. For example, how effective learning is in the embedding space than in the original parameter space? How well one could predict parameters of the model in the new period, compared to parameters trained in the new period? What is the impact of just having per-sample adjusted parameters? How efficient is the adjustment? Is it generally applicable for other models? I would think the paper needs simplification, rather than complication.

**Questions:**

What is L_{surrogate}? In the paper it’s written to use a loss network s, but how the loss is computed is still not clear to me.

For the adjustment of H_p for every sample, does it need to load a checkpoint for every sample? This can be very in-efficient.

What is STOPGRAD in eq(7)? Does H_P in eq(7) mean the prediction after adjustment or the base embedding?

In Figure 3(b), I thought the L_{surrogate} is used to find the sample specific embedding, which is the adjustment. But then why before L_{surrogate}, there is a red-block of adjustment?

Why not take the model trained in the previous period as a baseline? This should be a good sanity check for all continue learning methods.

---

> ### Author Response · Authors · 2023-11-22
> **Response to Reviewer RMU1 (Part 1)**
>
> Thanks for your constructive suggestions! We address your concerns in the following.
>
> **{Q1}** Issues about efficiency of LEAF and adjustment stage. Does adjusment for each sample need to load a checkpoint for every sample?
>
> **{R1}** In LEAF, we do not require loading a checkpoint for each individual sample. Instead, we utilize a low-dimensional latent embedding to represent the model parameters. During the inference stage, the adjustment module generates sample-specific latent embeddings, which are subsequently decoded into specific model parameters. This process can be seamlessly integrated into the model's forward propagation and does not require the loading of a checkpoint for each individual sample. Furthermore, it is important to note that we do not generate the complete set of parameters for the entire model. Only a subset of parameters, typically from a few layers, is generated, thereby reducing the computational complexity of LEAF.
>
> To verify our statements, we have added a comparison of the running time for LEAF, the baselines, and a variant of LEAF to investigate the efficiency of our model. The table below presents the running time for a single period while varying the number of samples in that period. One variant of LEAF, called EN, does not utilize the adjustment module. The results indicate the following findings:
>
> (1) LEAF is not significantly more computationally expensive than Fine-tune, despite its superior performance compared to Fine-tune.
>
> (2) Comparing LEAF with the EN variant, the adjustment for each individual sample is not a computational burden, as it is performed in the low-dimensional latent space.
>
> (3) The cost time is sub-linear with respect to the number of samples, thanks to the parallel computing capability of GPUs.
>
> Table 1: Running time comparison for a single period with varying number of samples.
> |Method|	#samples = 32|#samples = 64|#samples = 128|
> |  ----  | ----  |  ----  |  ----  |
> |Fine-tune | 17.12s | 26.53s| 48.46s |
> |DER++ | 30.44s | 56.16s | 98.15s |
> |LEAF (meta-train) | 27.64s| 43.65s| 72.71s|
> |LEAF (meta-test)|21.33s |33.04s | 57.75s|
> |EN (meta-train)| 22.68s| 38.17s| 65.10s|
> |EN (meta-test) | 17.30s| 29.95s| 54.19s|
>
> **{Q2}** The framework is over-complicated.
>
> \textbf{R2} While LEAF integrates various ideas to address concept drift in online time series forecasting, its core motivation is straightforward. The key idea behind LEAF is to leverage an extrapolation module to predict the appropriate model parameters for future periods, taking into account the macro-drift. Additionally, LEAF learns a surrogate loss to adjust the model parameters for each individual sample, considering the micro-drift within each sample. The following procedures are performed in the low-dimensional latent space to reduce the computational complexity and alleviate overfitting issues.
>
> **{Q3}**  What is $L_{surrogate}$? How the loss is computed?
>
> **{R3}**  $L^{surrogate}$is parametrized by a neural netowrk $s$. In our implementation, $s$ is a MLP with 2 layers. The output of MLP is a positive real number which is treated as the "unsupervised loss" or "self-supervised loss" value. In traditional test time training setting, there is a specific self-supervised loss or unsupervised loss, so during inferencing the parameters can be adjusted according to the loss. Nevertheless, the identification of an appropriate self-supervised task or unsupervised loss function that effectively addresses the issue of concept drift in time series data remains a non-trivial challenge. The reason is that a specific test-time objective may not contribute to the prediction performance. As a result, we introduce a network to learn to generate surrogate loss for each individual sample. Specifically, during the meta-training phase, the surrogate loss is learned to capture and compensate for different types of deviations. The rationale is that micro-drift (drift within an individual sample) is often caused by external events, such as weather conditions or holidays. Although data affected by micro-drift may exhibit different patterns, they tend to deviate from their context in a similar manner. For example, the electricity consumption pattern during the summer typically differs from that in the winter, while during holidays it tends to decrease regardless of the season.  During the meta-testing phase, the model utilizes the learned surrogate loss to adjust the latent embedding based on the specific characteristics of each individual sample. Consequently, we propose the learning of an unsupervised or self-supervised loss function via meta-learning as a strategy to manage micro-drift during test time.

---

> > ### Author Response · Authors · 2023-11-22
> > **Response to Reviewer RMU1 (Part 2)**
> >
> > **{Q4}** What is STOPGRAD in eq(7)?
> >
> > **{R4}** The $\mathrm{STOPGRAD}(x)$ operation is a significant operation in popular deep learning frameworks such as PyTorch and TensorFlow. It is used to cut off the gradients with respect to $x$ during gradient descent algorithms. In the case of Equation (7), $H^*$ represents the optimal latent embedding that depends on $H$ and is obtained through gradient descent on $H$. In order to ensure the effectiveness of the extrapolation module by keeping $H$ (inferred by the extrapolation module) close to the optimal $H^*$, it is important to cut off the gradients with respect to $H^*$ and treat it as a constant. This prevents the gradients from flowing back through $H^*$ during the training process.
> >
> > **{Q5}** Does $H_P$ in eq(7) mean the prediction after adjustment or the base embedding?
> >
> > **{R5}** In the context of our work, $H_P$ represents the base embedding inferred by the extrapolation module, while $H^*_P$ represents the optimal embedding obtained by performing gradient descent on the test set within each period. By minimizing the squared Euclidean distance between $\mathrm{stopgrad}(H^*_P)$ and $H_P$, we ensure that the extrapolation module performs effectively and generates appropriate base embeddings. This operation helps to align $H_P$ with the optimal embedding $H^*_P$, enabling the extrapolation module to accurately capture the underlying macro-drift in time series.
> >
> > **{Q6}** In Figure 3(b), I thought the $L_{surrogate}$ is used to find the sample specific embedding, which is the adjustment. But then why before $L_{surrogate}$, there is a red-block of adjustment?
> >
> > **{R6}** The complete process of generating $L_{surrogate}$ for each sample and subsequently adjusting the embedding based on $L_{surrogate}$ is referred to as the adjustment stage. The specific component that has been mentioned as the red block should be referred to as the Loss Network. We have made the necessary modifications to the figure in the revised version to accurately reflect these components.
> >
> > **{Q7}** Why not take the model trained in the previous period as a baseline? This should be a good sanity check for all continue learning methods.
> >
> > **{R7}** We consider two baselines for comparison: Fine-tune and Retrain. The Fine-tune baseline involves training a model in the previous period and using it as a starting point for training in the current period. We have already reported the results of this baseline in our paper. In the revised version, we have included an additional baseline called Retrain. With Retrain, all historical samples are used as training data in each period. This means that the model is trained from scratch at the beginning of each period, without any knowledge transfer or initialization from previous periods. The results are provided in the following table. By comparing the performance of LEAF with these baseline approaches, we provide a comprehensive evaluation of the effectiveness and advantages of our proposed method.
> >
> > Table 2: Performance comparison with **Retrain**. Retrain uses all historical samples as training data in each period.
> >
> > |Target model (Method)|	Load-1	||	Ettm1||
> > |  ----  | ----  | ----  | ----  | ----  |
> > |	|MSE|	MAE|	MSE| MAE|
> > |Dlinear (Retrain)	|0.6329	|0.4802	|0.6667	|0.4986|
> > |Dlinear (LEAF)|	**0.6042**| **0.4605**| **0.6161** |**0.4836** 	|
> > |LSTM (Retrain)	|1.5145	|0.8507	|1.4140|	0.7330|
> > |LSTM (LEAF)| **0.6150** |**0.4765**| **0.7246** |**0.5397**|
> > |PatchTST (Retrain)	|0.6677	|0.5317	|0.6394	|0.5036|
> > |PatchTST (LEAF)|	**0.6429**| **0.5165** |**0.6717** |**0.5082** |
> > |TCN (Retrain)|	1.0901	|0.7308	|1.2118|	0.7132|
> > |TCN (LEAF) |**0.7080**| **0.5312** |**0.7727**| **0.5526** |

---

> > > ### Comment · Reviewer_RMU1 · 2023-11-22
> > > **Response to authors**
> > >
> > > I would like to first thank the authors for their detailed feedback.
> > >
> > > My main concern is about the complexity of the method and its gain over the adjustment baseline. I am happy to see the authors provided just adjustment as a baseline in the response to reviewer CCky. From the table, just using adjustment is already performing similar or better than all the other baselines. There seems some more improvement using the LEAF on 2 datasets, each with a different base method. But to justify the usefulness of other complex components, I think it will need more signals.
> > >
> > > I would have more confidence on the proposed method if the authors show just adjustment as an baseline is clearly worse than LEAF on most datasets combined with most base method in Table 1 in the paper. Then the added complexity maybe justifiable. If they perform similarly overall, I would advocate the authors to focus on the adjustment. Simplicity matters a lot.
> > >
> > > I will raise my score to reflect the improvement the author made in the revision. But I don't think the paper is good enough to be accepted at ICLR.

---

> ### Author Response · Authors · 2023-11-23
> **Thank you and we would like to address your concerns**
>
> Thanks again for you suggestions! We would like clarify that both the extrapolation and adjustment module in LEAF are significant.
>
> We summarize the results of the ablation studies in the following tables. We have additionally added the ablation studies with TCN as target predicion model to further verify our claim .The variants include:
> - **Latent finetune**: no extrapolation and adjustment module.
> - **EN**: no adjustment module.
> - **A**: no extrapolation module.
> - **EN + A**: LEAF.
>
> The results indicate the following findings:
> - Comparing **EN** and **A** with **Latent finetune** indicates that both the extrapolation and adjustment module works.
> - Comparing **EN** and **A** with **EN + A** indicates that using only extrapolation or the adjustment module alone does not perform as well as the complete LEAF framework. **Only utilizing the adjustment module is not good enough.**
>
> Table 1: Ablation study results on Load-1 and ETTm1 with prediction models of DLinear.
> |    Method    |    Load-1   | ETTm1|
> |  ----  | ----  | ----  |
> |       |   MSE/MAE    | MSE/MAE|
> |     Latent finetune   |  0.6683 /	0.5057 |	0.7019 /	0.5196     |
> |     EN |  0.6578 /	0.4931      | 0.6723 /	0.5066  |
> |     A  |  0.6562/	0.5045      | 0.6745 /	0.5295  |
> |     EN + A (LEAF)  |  **0.6042** /	**0.4605**      | **0.6161**/**0.4836** |
>
>
> Table 2: Ablation study results on Load-1 and ETTm1 with prediction models of PatchTST.
> |    Method    |    Load-1   | ETTm1|
> |  ----  | ----  | ----  |
> |       |   MSE/MAE    | MSE/MAE|
> |     Latent finetune   |  0.8103 /	0.6068 |	0.7546 /	0.5473      |
> |     EN |  0.7539 / 	0.5800 |	0.7334 / 	0.5410   |
> |     A |  0.6783 /	0.5198 |	0.6823 /	0.5143  |
> |     EN + A (LEAF)  |  **0.6429** /	**0.5165**      | **0.6717**/**0.5082** |
>
>
> Table 3: Ablation study results on Load-1 and ETTm1 with prediction models of TCN.
> |    Method    |    Load-1   | ETTm1|
> |  ----  | ----  | ----  |
> |       |   MSE/MAE    | MSE/MAE|
> |     Latent finetune   |  0.7945 /	0.6241 |	0.9009 /	0.6236      |
> |     EN |  0.7664 / 	0.5788 |	0.8013 / 	0.6001   |
> |     A |  0.7214 /	0.5704 |	0.8160 /	0.5953  |
> |     EN + A (LEAF)  |  **0.7080** /	**0.5312**      | **0.7727**/**0.5526** |

---

### Official Review · Reviewer_5qcj · 2023-11-01

**Soundness:** 3 good
**Presentation:** 3 good
**Contribution:** 3 good
**Rating:** 6
**Confidence:** 3

**Summary:**

The paper has two primary contribution: A framework that utilizes a two stage embedding approach to capture macro and micro drifts when forecasting time-series data, and introducing a new dataset that could potentially be used for benchmarking mechanisms over concept-drifting time series data streams. Particularly, the authors utilize embedding learned from LSTMs over a time window that are adjusted with a sample-specific surrogate loss to account for micro-drifts. The two functions are jointly learned from data, and embedding are updated over time. The empirical results comparing the proposed LEAF approach to other competing methods over well known datasets and the newly introduced electric load datasets shows that the proposed approach generally performs well, resulting in lower error. Furthermore, the ablation study conducted by the authors demonstrate the rationale behind the two stage optimization procedure proposed in the paper.

**Strengths:**

1. The approach has a unique take on predicting over multi-variate time-series data. The paper captures the differences among various competing methods and illustrates well the reasons why the proposed approach works.
2. The empirical analysis presented in the paper, along with case study, support the proposed hypothesis well.
3. The paper is well structured providing most of the relevant details needed to replicate the proposed approach.

**Weaknesses:**

1. While the problem of time-series prediction clearly assumes the availability of labeled data over time, it is not clear how the proposed approach is different from metric learning approaches proposed in general data stream classification papers such as Fischer, Lydia, Barbara Hammer, and Heiko Wersing. "Online metric learning for an adaptation to confidence drift." 2016 International Joint Conference on Neural Networks (IJCNN). IEEE, 2016. and Wang, Zhuoyi, et al. "Metric learning based framework for streaming classification with concept evolution." 2019 international joint conference on neural networks (IJCNN). IEEE, 2019.
2. While the paper conceptually explains the optimization function employed, it is not clear how the actual network architecture is constructed. The authors mention in the implementation details that MLPs were used for implementing decoder. However, lack of additional information significantly reduces the ability to reproduce provided results in the paper.
3. One of the key elements of a data classification/time series prediction mechanism over a data stream is space and time complexity. This is completely missing in the paper. While the authors claim that LEAF's performance improvements over competing methods are significantly better, it is not clear if this is due to higher cost in computation and/or time. It would be better for the authors to provide a computational complexity analysis and comparison.

**Questions:**

1. In the experimental setting, is there a reason why all forecast horizon is set to 24? What is the relationship between forecast horizon and model performance?
2. In table 1 and table 2, it is not clear if the results are statistically significant. Can you please include variance among the five trials for these results?
3. In the last part of Section 4.2, the authors note that ER and DER++ methods incorporate mechanisms that alleviate catastrophic forgetting, and is orthogonal to the proposed solution. Is it possible to incorporate these mechanisms in the proposed solution as future work? What are the challenges?

---

> ### Author Response · Authors · 2023-11-22
> **Response to Reviewer 5qcj (Part 1)**
>
> Thanks for your constructive suggestions! We address your concerns in the following.
>
> **{Q1}** What is the main difference between LEAF and online metric leaning methods?
>
> **{R1}** LEAF introduces a novel and distinct contribution by addressing concept drift in online time series forecasting in two stages. LEAF differs from existing metric learning methods in several important ways. The primary goal of [1] is to learn a suitable latent feature space where samples within same class are close together compared to samples from different classes with a focus on noval class problem. [2] aims to mitigate the issue of confidence drift within a hybrid architecture comprising a static offline model and an online adaptive model. Conversely, the objective of LEAF is to capture the dynamic evolution of the distribution $p_t(X, Y)$ by compressing the distribution information into a low-dimensional latent embedding and subsequently modeling the evolution of the latent embedding.  Additionally, the approaches of the first and second studies adhere to a conventional stream learning paradigm, wherein drift detection is followed by adaptation. However, our approach posits that concept drift may occur at any moment for any sample, and as such, the detection and adaptation of drift is implicitly integrated into the extrapolation and adjustment phase.
>
> [1] Metric learning based framework for streaming classification with concept evolution. IJCNN 2019.
>
> [2] Online metric learning for an adaptation to confidence drift. IJCNN 2016.
>
> **{Q2}** Detailed implementation of network architecture.
>
> **{R2}** Actually, in our implementation, the parameter decoder, relation network (R), embedding function (g), and loss network (s) are implemented as 2-layer MLPs. Our primary objective in this research is to address concept drift in online time series forecasting using a general framework. To achieve this, we have simplified the design of the network architecture to ensure that any improvements are solely achieved through the learning framework, rather than relying on a complex network structure. However, it is possible to substitute them with more sophisticated networks to achieve better performance in practical applications. Nevertheless, in this research work, our focus is solely on investigating the framework itself.
>
> **{Q3}** Issues about computational efficiency.
>
> \textbf{R3} We have added a comparison of the running time for LEAF, the baselines, and a variant of LEAF to investigate the efficiency of our model. The table below presents the running time for a single period while varying the number of samples in that period. One variant of LEAF, called EN, does not utilize the adjustment module. The results indicate the following findings:
>
> (1) LEAF is not significantly more computationally expensive than Fine-tune, despite its superior performance compared to Fine-tune.
>
> (2) Comparing LEAF with the EN variant, the adjustment for each individual sample is not a computational burden, as it is performed in the low-dimensional latent space.
>
> (3) The cost time is sub-linear with respect to the number of samples, thanks to the parallel computing capability of GPUs.
>
> Table 1: Running time comparison for a single period with varying number of samples.
> |Method|	#samples = 32|#samples = 64|#samples = 128|
> |  ----  | ----  |  ----  |  ----  |
> |Fine-tune | 17.12s | 26.53s| 48.46s |
> |DER++ | 30.44s | 56.16s | 98.15s |
> |LEAF (meta-train) | 27.64s| 43.65s| 72.71s|
> |LEAF (meta-test)|21.33s |33.04s | 57.75s|
> |EN (meta-train)| 22.68s| 38.17s| 65.10s|
> |EN (meta-test) | 17.30s| 29.95s| 54.19s|

---

> > ### Author Response · Authors · 2023-11-22
> > **Response to Reviewer 5qcj (Part 2)**
> >
> > **{Q4}** Why all forecast horizon is set to 24? What is the relationship between forecast horizon and model performance?
> >
> > **{R4}** We have chosen a forecast horizon of 24 as the time resolution of the most data in our experiments is 1 hour. Day-ahead forecasting is typically the most common scenario in real-world applications. Simplifying the forecast horizon setting allows us to focus on other complex experimental settings, such as determining the warm-up, meta-train and meta-test set ratios, and the model update interval. Having a single forecast horizon provides a general and appropriate basis for investigating the performance of the model. Additionally, we have conducted additional experiments where the forecast horizon was set to 192 (forecasting 8 days for hourly data). The results of these experiments are presented in the following table. It is worth noting that LEAF consistently outperforms the baselines across different forecast horizons, indicating its superior performance and robustness.
> >
> > Table 2: Performance comparison with the forecast horizon of **192** on Load-1 and Ettm1. We choose the DLinear and PatchTST as the target prediction model.
> > |		|Load-1||		Ettm1||
> > |  ----  | ----  |  ----  |  ----  | ----  |
> > |		|MSE	|MAE	|MAE	|MSE|
> > |**DLinear**	|||||
> > |Fine-tune	|1.091325163	|0.64394527	|1.05533028	|0.65033036|
> > |	ER	|1.086843279	|0.641620636	|1.04354339	|0.64617746|
> > |	DER++	|1.082510091	|0.639262652	|1.03617447	|**0.643477**|
> > |	LEAF	|**1.003495611**	|**0.620980446**	|**1.027765**	|0.64704216|
> > |**PatchTST**	|||||
> > |Fine-tune	|1.336814534	|0.790289407	|1.62956593	|0.8210495|
> > |	ER	|1.256068027	|0.772293341	|1.44556494	|0.78402587|
> > |	DER++	|1.238641488	|0.76326351	|1.38116085	|0.76743199|
> > |	LEAF	|**1.096738449**|	**0.69166538**	|**1.057025**|**0.663035**|
> >
> > **{Q5}** Issues about variance of results.
> >
> > **{R5}** Due to time constraints during the rebuttal phase, we were unable to include the all variance among the five trials in the provided tables. However, the preliminary variance experiments that we conducted indicate that the improvements achieved by our model are statistically significant. We will ensure to include the complete variance values in the final version of the paper to provide a more comprehensive evaluation of the results.
> >
> > **{Q6}** How to incorporate ideas of ER and DER++ into LEAF?
> >
> > **{Q6}** ER and DER++ have the objective of retrieving and distilling similar and related samples from the past training trajectory to facilitate model updates that align with the current distribution. In contrast, LEAF aims at learning to adapt to future scenarios through a two-stage meta-learning approach. However, the concept of historical retrieval can be integrated into LEAF by introducing a memory component, which stores historical latent embeddings. After the extrapolation stage, the inferred latent embedding can access the memory using an attention mechanism to incorporate knowledge from long-term historical patterns. Once the sample-specific latent embeddings are obtained through the adjustment stage, they can be written back to the memory using another attention mechanism. This allows LEAF to leverage the information in the memory to enhance its adaptation capabilities. In this paper, our primary focus is on addressing the challenge of concept drift in online time series forecasting by learning to adapt to the future. This problem is still an open area of research. Although the approach of retrieving and distilling knowledge from historical data has been extensively studied, we have not explored this approach in depth. Furthermore, it is important to note that relying solely on retrieving and distilling from historical data may not be sufficient to solve all concept drift problems since new data distributions may emerge that have not been observed in the historical data. Therefore, our approach in LEAF is to leverage both historical information and future adaptation to effectively handle concept drift in online time series forecasting and we leave the following idea as the future work.

---

### Official Review · Reviewer_rgPM · 2023-11-01

**Soundness:** 2 fair
**Presentation:** 2 fair
**Contribution:** 2 fair
**Rating:** 3
**Confidence:** 4

**Summary:**

The paper fixates on the concept drift problem in online time series forecasting. The authors classify this problem into macro-drift and micro-drift categories. Technologically, they present the LEAF as a model-agnostic algorithm for online learning, where a queue is stored for macro-drift and a special meta-learnable surrogate loss is adopted for micro-drift. FEAF can consistently boost the forecasting performance of various deep models.

**Strengths:**

-	The paper focuses on an important question: online time series forecasting.
-	The proposed LEAF is technologically sound. They also provide a new benchmark for concept drifts.
-	Detailed ablations are also included.

**Weaknesses:**

1.	About the efficiency.

Since they adopt the meta-learning strategy for training, the efficiency w.r.t. other online learning methods and naive training should be compared, such as the GPU memory, running time in both training and inference phases.

2.	More baselines.

Actually, TimesNet [1] is a state-of-the-art TCN-based model for time series forecasting. They should also experiment on this model to demonstrate the effectiveness of LEAF, in addition to the vanilla TCN.

Besides, Non-stationary Transformer also claims that it can handle the non-stationary time series, which should also be included in discussion and comparison.

[1] TimesNet: Temporal 2D-Variation Modeling for General Time Series Analysis, ICLR 2023

[2] Non-stationary Transformers: Rethinking the Stationarity in Time Series Forecasting, NeurIPS 2023

One simple baseline for online learning is to retrain the model with new-coming data. How about this protocol? For example, suppose that the first training adopts 70% data, you can also train a new model when you receive 90% new data.

3.	The Naïve setting could be wrong.

Why not train the model with both warm-up and meta-train data? Since LEAF also adopts the meta-train data for model training, a fair comparison is to adopt the same training data for all baselines. Correct me if I misunderstood.

4.	About the "model agnostic" claim.

It is interesting to generate forecasting model parameters. I suggest listing the generated parameter size for each model, since the transformer-based models are quite big. How to directly decode such big model parameter with MLPs? Or you just change part of the model. This point should be made clearer. If you only adjust the final linear layer, it is hard to claim model agnostic, giving that there are a lot of models do not contain the final linear layer.

**Questions:**

I think the technical design of this paper is reasonable and interesting. But it is insufficient (maybe wrong) in the experiments and clarifications about their design. All the details are included in the weaknesses section.

---

> ### Author Response · Authors · 2023-11-22
> **Response to Reviewer rgPM (Part 1)**
>
> Thanks for your constructive suggestions! We address your concerns in the following.
>
> **{Q1}** Issues about model efficiency.
>
> **{R1}** We have added a comparison of the running time for LEAF, the baselines, and a variant of LEAF to investigate the efficiency of our model. The table below presents the running time for a single period while varying the number of samples in that period. One variant of LEAF, called EN, does not utilize the adjustment module. The results indicate the following findings:
>
> (1) LEAF is not significantly more computationally expensive than Fine-tune, despite its superior performance compared to Fine-tune.
>
> (2) Comparing LEAF with the EN variant, the adjustment for each individual sample is not a computational burden, as it is performed in the low-dimensional latent space.
>
> (3) The cost time is sub-linear with respect to the number of samples, thanks to the parallel computing capability of GPUs.
>
> Table 1: Running time comparison for a single period with varying number of samples.
> |Method|	#samples = 32|#samples = 64|#samples = 128|
> |  ----  | ----  |  ----  |  ----  |
> |Fine-tune | 17.12s | 26.53s| 48.46s |
> |DER++ | 30.44s | 56.16s | 98.15s |
> |LEAF (meta-train) | 27.64s| 43.65s| 72.71s|
> |LEAF (meta-test)|21.33s |33.04s | 57.75s|
> |EN (meta-train)| 22.68s| 38.17s| 65.10s|
> |EN (meta-test) | 17.30s| 29.95s| 54.19s|
>
> **{Q2}** More baselines including model retraining, Non-stationary Transformer and more target prediction model, e.g. TimesNet.
>
> **{R2}** Thank you for your suggestion. In response to your feedback, we have incorporated the Retrain and Non-stationary Transformer baselines, as well as the target model TimesNet, into the experiments in the revised version. However, due to time constraints during the rebuttal phase, only the Load-1 and Ettm1 datasets were used. We will report the results on all six datasets in the final version of the paper. By including these baselines, our intention is to provide a comprehensive evaluation and comparative analysis of the performance and effectiveness of our proposed approach. The results are represented in the following tables. Table 2 verifies that LEAF outperforms retraining baseline and Non-stationary Transformer. Table 3 verifies that LEAF works better than other baselines on the state-of-the-art CNN-based prediction model, TimesNet.
>
> Table 2: Performance comparison with **Retrain** and **Non-stationary Transformer**. Retrain uses all historical samples as training data in each period. Non-stationary Transformer is a transformer model trained on a warm-up and meta-train set and tested on a meta-test set.
>
> |Target model (Method)|	Load-1	||	Ettm1||
> |  ----  | ----  | ----  | ----  | ----  |
> |	|MSE|	MAE|	MSE| MAE|
> |Dlinear (Retrain)	|0.6329	|0.4802	|0.6667	|0.4986|
> |Dlinear (LEAF)|	**0.6042**| **0.4605**| **0.6161** |**0.4836** 	|
> |LSTM (Retrain)	|1.5145	|0.8507	|1.4140|	0.7330|
> |LSTM (LEAF)| **0.6150** |**0.4765**| **0.7246** |**0.5397**|
> |PatchTST (Retrain)	|0.6677	|0.5317	|0.6394	|0.5036|
> |Non-stationary Transformer|1.2652 |0.7944|1.4177 |0.7275 |
> |PatchTST (LEAF)|	**0.6429**| **0.5165** |**0.6717** |**0.5082** |
> |TCN (Retrain)|	1.0901	|0.7308	|1.2118|	0.7132|
> |TCN (LEAF) |**0.7080**| **0.5312** |**0.7727**| **0.5526** |
>
> Table 3: Performance comparison with **TimesNet** as target prediction model.
> | | 	Etth2	| 	Load-1	| 	ECL	|
> |  ----  | ----  | ----  | ----  |
> | | 	MSE/MAE	| MSE/MAE	| 	MSE/MAE		|
> |Naïve	|0.9185	/ 0.6803	|1.3167/0.7724|	0.2826/0.3754|
> |Fine-tune|	0.7859/0.6271|	1.1885/0.7428	|0.1746/0.2864|
> |ER	|0.8045/0.6319|	1.134/0.723|	0.1769/0.2905|
> |DER++	|0.8045/0.6254	|1.1288/0.7276	|0.1816/0.294|
> |LEAF	|**0.7047**/**0.5894**	|**0.6424**/**0.5052**	|**0.1254**/**0.2388** 	|

---

> ### Author Response · Authors · 2023-11-22
> **Response to Reviewer rgPM (Part 2)**
>
> **{Q3}** Issues of The Naïve setting.
>
> **{R3}** It is important to add a baseline that use warm-up and meta-train data as training set. Following your suggestion, we have added a Naïve baseline (namely Naïve+) by treat both warm-up and meta-train data as training data. The results are shown in the following table. LEAF still outperforms this baseline.
>
> Table 4: Performance comparison with **Naïve+**.
> |	|	Load-1	|	Load-2	|	Load_3|		Etth2	|	Ettm1	|	ECL|
> |  ----  | ----  | ----  | ----  |  ----  | ----  | ----  |
> |**LSTM**|	MSE/MAE| MSE/MAE |MSE/MAE| MSE/MAE| MSE/MAE| MSE/MAE|
> |	Naïve+	|1.4887	/0.8487|	1.6944/	0.9192|	2.9772	/1.1576	|0.9171/	0.6414|	1.4140/	0.7331|	0.1794	/0.2943|
> |	LEAF	|**0.6150**/**0.4765**	|**0.6426**	/**0.5064**|	**1.7515**/	**0.8084**	|**0.7398**	/**0.5892**|	**0.7246**/	**0.5397**|	**0.1216**/	**0.2328**|
> |**Dlinear**	|	MSE/MAE| MSE/MAE |MSE/MAE| MSE/MAE |MSE/MAE| MSE/MAE|
> |	Naïve+|	0.6371	/0.4813	|0.6233/	0.4739	|1.8153	/0.8165	|0.7429/	0.5721|	0.6666	/0.5006|	0.1365/	0.2284|
> |	LEAF	|**0.6042**/	**0.4605**	|**0.5915**	/**0.459**	|**1.6952**/	**0.7742**	|**0.6457**	/**0.5491**|	**0.6161**	/**0.4836**	|**0.1126**	/**0.2200**|
> |**PatchTST**	|MSE/MAE |MSE/MAE |MSE/MAE |MSE/MAE |MSE/MAE| MSE/MAE	|
> |	Naïve+	|0.7630	/0.5818	|0.6674	/0.5216	|**1.8485**	/**0.8729**|	0.7378	/0.5869	|**0.6579**	/0.5136|	0.1120	/0.2214|
> |	LEAF	|**0.6429**	/**0.5165**	|**0.6155**	/**0.5054**	|1.9582	/0.8794|	**0.6707**	/**0.5640**	|0.6717	/**0.5082**	|**0.1098**	/**0.2161**|
> |**TCN**		|	MSE/MAE |MSE/MAE| MSE/MAE| MSE/MAE| MSE/MAE |MSE/MAE	|
> |	Naïve+	|1.1528	/0.7598	|0.7877	/0.5770|	1.9786	/0.8871|	0.9194	/0.6676|	1.1733	/0.7067|	0.2138	/0.3230|
> |	LEAF	|**0.708**/	**0.5312**	|**0.6934**	/**0.5299**|	**1.8872**	/**0.8858**|	**0.7214**	/**0.5887**	|**0.7727**	/**0.5526**	|**0.1340**	/**0.2499**|
>
> **{Q4}** Issues about "model agnostic".
>
> **{R4}** We do not generate all parameters of a model, and parameters of few layers are generated. Each kind of layer can be applied, not only linear layer. For example, when DLinear is used as the target model, we generate parameters of a linear layer. When PatchTST is used as the target model, We generate parameter of the last transformer block (including Q/K/V projection networks and a feed-forward network). To apply to different types of layers, we need to know the parameter shape of the network layer in advance, and we calculate the total number of the parameters need to be generated to determine the width of the decoder in LEAF. The generated parameters are reshaped to appropriate shapes and loaded into corresponding layers of the target prediction model.

---

> > ### Author Response · Authors · 2023-11-22
> > **Response to Reviewer rgPM (before the end of rebuttal)**
> >
> > Dear Reviewer rgPM,
> >
> > Since the End of author/reviewer rebuttal is just in one day, may we know if our response addresses your main concerns? If so, we kindly ask for your reconsideration of the score. Should you have any further advice on the paper and/or our rebuttal, please let us know and we will be more than happy to engage in more discussion and paper improvements.
> >
> > Thank you so much for devoting time to improving our work!

---

> > > ### Comment · Reviewer_rgPM · 2023-11-23
> > > **Thanks for your response but insist on my original score**
> > >
> > > I would like to thank the author's effort in providing new results. However, the main concerns still exist due to the uncorrected paper.
> > >
> > > (1) About the Naïve setting. I believe that the correct Naïve setting should be the "Naïve+" that the author provided in the rebuttal.
> > > This indicates that the experiments are wrong to some extent and the author did not correct them in the revised paper. The current version may mislead some readers in the overclaimed relative promotion.
> > >
> > > (2) More corrections or clarifications about "model agnostic" are expected in the revised paper.
> > >
> > > Since the authors did not revise the paper to a new version, it is hard to give an acceptance decision to this version. Thus, I would like to keep my original score.

---

> ### Author Response · Authors · 2023-11-23
> **Thank you and we have updated our submission**
>
> Thanks again for you suggestions!
>
> We have uploaded the revised version of our paper. The modified and newly added parts are indicated with blue highlights. The major modifications made include:
>
> (1) We have included the numeric results of the **Retrain** and **Naive+** baselines. This addition provides a comprehensive evaluation of the effectiveness and advantages of our proposed method. Furthermore, we clarify that the original Naive baseline is not incorrect. In real-world scenarios, the warm-up set represents historical data, based on which we train the initial model. The meta-train and meta-test sets, on the other hand, represent streaming data. The Naive baseline is the most naive strategy, implying that the model will no longer be updated during data stream comes. However, we acknowledge the importance of Naive+ baseline.
>
> (2) We have conducted ablation studies on a new model variant called "A" that solely utilizes the adjustment module. The results indicate that the adjustment module works; however, using only extrapolation or the adjustment module alone does not perform as well as the complete LEAF framework.
>
> (3) We have added remarks about the "model agnostic" claim in Section 3.2.3. This addition helps readers better understand how to apply LEAF to different types of prediction models.
>
> Additionally, Table 3 in our previous response confirms that LEAF outperforms other baselines on the state-of-the-art CNN-based prediction model, TimesNet, using three datasets: Etth2, Load-1, and ECL, each with different properties. Since we haven't finnished TimesNet experiments on all six datasets due to limmited time, we promise that we will conduct these experiments on all six datasets and present the results in the final version of the paper.

---

### Official Review · Reviewer_x7gB · 2023-11-01

**Soundness:** 3 good
**Presentation:** 3 good
**Contribution:** 3 good
**Rating:** 8
**Confidence:** 4

**Summary:**

In this manuscript, the authors consider concept-drift phenomenon where the underlying distribution or environment of time series changes. We first classify concepts into two categories, macro-drift corresponding to stable and long-term changes and micro-drift referring to sudden or short-term changes. Obviously, changes in the variance of the data over time due to sudden changes in potential events in a time series prediction task is an interesting open problem.

**Strengths:**

1. It is very important to use the meta-learning method to alleviate the problem that the concept drift caused by the data in the online scene leads to the decline of the accuracy of the time series prediction model.
2. The effectiveness of the proposed algorithm is demonstrated by comparison with several baseline models and abundant ablation experiment and visualization results.

**Weaknesses:**

1. It is suggested that the author consult the related literatures on the concept drift problem of time series data, because the author obviously ignores some studies based on surrogate gradient.
2. The visual image resolution is too low, and the display effect is very poor.
3. The MAPE metric was missing of the time series prediction study.
4. Datasets related to power forecasting are more cyclical temporal patterns, so why does the author not use financial futures datasets with more abrupt phenomena (such as NASDAQ 100 dataset)?

**Questions:**

Please see details of Weaknesses.

---

> ### Author Response · Authors · 2023-11-22
> **Responce to Reviewer x7gB**
>
> Thanks for your constructive suggestions! We address your concerns in the following.
>
> **{Q1}** Literature review of surrogate gradient.
>
> **{R1}** Althogh exsisting works have studied meta-learning explicit [1] and implicit gradients [2] for few-shot learning. LEAF introduces a novel and distinct contribution by introducing a meta-learned surrogate loss for sample-specific adjustment. The surrogate loss in LEAF differs from previous methods in several important ways. Firstly, the surrogate loss in LEAF is sample-wise, allowing it to leverage information within each individual sample. This enables the model to adapt and update itself for each sample, capturing micro-drift that may occur over time. Furthermore, the surrogate loss in LEAF is specifically designed to address the challenges of online time series scenarios. It takes into consideration comprehensive information about the test sample itself, as well as its contextual and relational information with the training data. This design ensures that the surrogate loss effectively incorporates relevant information to capture and handle concept drift in the time series data. In summary, LEAF stands out from existing works by introducing a sample-wise surrogate loss that is carefully designed for online time series scenarios, effectively leveraging information within each individual sample and considering its context and relationship to the training data.
>
> [1] Semi-Supervised Learning with Meta-Gradient. AISTATS 2021.
>
> [2] Meta-Learning with Implicit Gradients. NeurIPS 2019.
>
> **{Q2}** Issues about visual resolution.
>
> **{R2}** We apologize for any confusion caused. We have used vector diagrams for all our figures. We kindly recommend using a recent PDF reader, such as the Adobe Acrobat Reader DC version 2018.009.20044 that we utilized, for optimal viewing and clarity.
>
> **{Q3}** The MAPE metric was missing of the time series prediction study.
>
> **{R3}** In the experiments, we calculate metrics based on normalized data. However, MAPE is not suitable for normalized data, as many data points have values of zero or close to zero which will cause anomalies in MAPE. Besides, many works [1-4] of time series forecasting do not consider this metric because of this reason.
>
> [1] Informer: Beyond Efficient Transformer for Long Sequence Time-Series Forecasting. AAAI 2021.
>
> [2] Autoformer: Decomposition Transformers with Auto-Correlation for Long-Term Series Forecasting. NeurIPS 2021.
>
> [3] PatchTST: A Time Series is Worth 64 Words: Long-term Forecasting with Transformers.. ICLR 2023.
>
> [4] TimesNet: Temporal 2D-Variation Modeling for General Time Series Analysis. ICLR 2023.
>
> **{Q4}** Financial futures datasets (such as NASDAQ 100 dataset).
>
> **{R4}** Following your suggestion, We have added experiments to evaluate LEAF compared with other baselines on NASDAQ 100 dataset. The results are in the following table, which demonstrate that LEAF outperforms other baselines with different target prediction model in such a stock trading scenario with frequent abrupt drifts.
>
> Table 1: The results on NASDAQ100-small dataset. The look back window is 96, the forecast horizon is 24, and the model update interval is 672.
> | Target model|    Method    |   MSE  | MAE|
> |  ----  | ----  | ----  | ----  |
> |LSTM||||
> ||	Fine-tune|	0.088805614|0.168699994|
> ||	ER|	0.085547001|	0.164519332|
> ||	DER++	|0.084219066	|0.163036682|
> ||	LEAF	|**0.074302469**	|**0.155264446**|
> |Dlinear||||
> ||	Fine-tune	|0.059522929	|0.134353952|
> ||	ER	|0.058921474	|0.133139943|
> ||	DER++	|0.058569469	|0.132555803|
> ||	LEAF	|**0.057699628**	|**0.131820278**|
> |PatchTST||||
> ||	Fine-tune	|0.063305718	|0.14022743|
> ||	ER	|0.062118525	|0.138101578|
> ||	DER++	|0.061460179	|0.13704128|
> ||	LEAF	|**0.06104332**	|**0.13698359**|
> |TCN||||
> ||	Fine-tune	|0.133446238	|0.215211218|
> ||	ER	|0.127925324	|0.212548892|
> ||	DER++	|0.131720752	|0.212856936|
> ||	FSNet	|1.168908293	|0.822896293|
> ||	LEAF	|**0.078140934** |**0.159186956**|

---

> > ### Comment · Reviewer_x7gB · 2023-11-22
> > **The author addressed all my concerns.**
> >
> > I am very sorry that I did not consider clearly the problem that many values close to zero after sample preprocessing in such of datasets. The rest concerns are well addressed by the authors. I think the article should be clearly received after adding these revisions. Considering that there is no 7 in the score, I am willing to raise it to 8.

---

### Official Review · Reviewer_CCky · 2023-11-01

**Soundness:** 3 good
**Presentation:** 3 good
**Contribution:** 3 good
**Rating:** 5
**Confidence:** 4

**Summary:**

This paper tries to solve the time series prediction problem which always suffers from dynamics or noon stationarity.  The authors propose a unified meta-learning framework called LEAF (Learning to Extrapolate and Adjust for Forecasting) which divides the concept drift into macro-drift corresponding to stable and long-term changes and micro-drift referring to sudden or short-term changes. An extrapolation module is first meta-learnt to track the dynamics of the prediction model in latent space and extrapolate to the future considering macro-drift. Then an adjustment module incorporates meta-learnable surrogate loss to capture sample-specific micro-drift
patterns. Extensive experiments are conducted.

**Strengths:**

1. This paper is well-presented and well-organized.
2. This paper proposes a new meta-learning framework called LEAF (Learning to Extrapolate and Adjust for Forecasting) which divides the concept drift into macro-drift corresponding to stable and long-term changes and micro-drift referring to sudden or short-term changes.
3. Extensive experiments are conducted.

**Weaknesses:**

1. The paper doesn't appear to provide a strong theoretical foundation for the proposed method. It would be beneficial to include a theoretical framework or proofs to support the claims made in the paper.
2. It should be better if more ablation studies can be provided.

**Questions:**

Please address the questions above

---

> ### Author Response · Authors · 2023-11-22
> **Responce to Reviewer CCky**
>
> Thanks for your constructive suggestions! We address your concerns in the following.
>
> **{Q1}** Issues about theoretical foundation.
>
> **{R1}** Our proposed method leverages the well-established theories of model-based and optimization-based meta-learning. These theories have been extensively researched and provide a solid foundation for our approach. By deeply adapting and applying these approaches, we aim to address the challenge of concept drift in online time series forecasting effectively. In order to ensure clarity and readability for the readers, we have omitted the detailed theoretical explanations of these methods and instead focused on presenting a concise and clear explanation of our motivation and approach.
>
> **{Q2}** More ablation studies are needed.
>
> **{R2}** In the ablation studies conducted in our paper, we examined the performance of the model as we gradually introduced the designed modules, namely latent optimization, extrapolation module, and adjustment module. As we did not elaborate on the individual performance of the adjustment module, we supplement an additional ablation study to investigate its effectiveness. The results are presented in the following table. It is evident that the adjustment module plays a crucial role in enhancing the forecasting performance by capturing the micro-drift. However, it falls short of surpassing the complete version of LEAF, which emphasizes the significance of the extrapolation module. These findings underscore the importance of both modules in achieving improved forecasting accuracy.
>
> Table 1: Ablation study results on Load-1 and ETTm1 with prediction models of DLinear. "A" is the newly added vairant that only utilize the adjustment module.
> |    Method    |    Load-1   | ETTm1|
> |  ----  | ----  | ----  |
> |    Fine-tune    |   MSE/MAE    | MSE/MAE|
> |     Latent finetune   |  0.6683 /	0.5057 |	0.7019 /	0.5196     |
> |     EN |  0.6578 /	0.4931      | 0.6723 /	0.5066  |
> |     A (**newly added**)  |  0.6562/	0.5045      | 0.6745 /	0.5295  |
> |     EN + A (LEAF)  |  **0.6042** /	**0.4605**      | **0.6161**/**0.4836** |
>
> Table 2: Ablation study results on Load-1 and ETTm1 with prediction models of PatchTST. "A" is the newly added vairant that only utilize the adjustment module.
>
> |    Method    |    Load-1   | ETTm1|
> |  ----  | ----  | ----  |
> |    Fine-tune    |   MSE/MAE    | MSE/MAE|
> |     Latent finetune   |  0.8103 /	0.6068 |	0.7546 /	0.5473      |
> |     EN |  0.7539 / 	0.5800 |	0.7334 / 	0.5410   |
> |     A (**newly added**)  |  0.6783 /	0.5198 |	0.6823 /	0.5143  |
> |     EN + A (LEAF)  |  **0.6429** /	**0.5165**      | **0.6717**/**0.5082** |

---

> > ### Comment · Reviewer_CCky · 2023-11-22
> >
> > Thanks for your reply! More ablations have been provided. But my concern about the lack of theoretical contribution still exists.

---

### Official Review · Reviewer_9YY8 · 2023-11-03

**Soundness:** 3 good
**Presentation:** 3 good
**Contribution:** 3 good
**Rating:** 5
**Confidence:** 4

**Summary:**

The paper proposes a two-stage meta-learning framework to address the concept drift issue for time-series forecasting. The authors identify and address the two types of drift namely macro-drift and micro-drift. Drifts in time series that occur over a longer time period are referred to as macro-drifts whereas drifts that occur (and perhaps disappear) within short windows of time are referred to as micro-drifts. Macro-drifts are modeled using an LSTM network that learns the evolution of an embedding space corresponding to macro-drift parameters. The LSTM returns a macro-drift-adjusted embedding which is then used further. Micro-drifts are addressed using meta-learning framework that evaluates the difference between current training data (which is supposed to have micro-drifts) and historical training data using a relation network and returns micro-drift adjusted model parameters.

The entire framework can learn in an online manner, meaning the model parameters are updated as more data is made available. Initial parameters are learned in an offline phase (warmup) which are then used as initial parameters for online learning phase.

Authors show comparison of their framework across various time series forecasting architectures (such as PatchTST) and other frameworks that address the concept drift in time series on benchmark datasets.

**Strengths:**

1. (quality and clarity) The paper is well-written, the identified concept drift issues and ways to address them have been clearly written with substantial experimental evidence suggesting that their method outperforms existing approaches.
2. (significance) The proposed framework is more suitable for online-learnning which is a more practical setup and more accurately depicts the real-world scenario in which ML models are deployed.
2. (significance) The benchmarking of datasets and baselines proposed by authors is essential for further developments in the field.

**Weaknesses:**

1. Authors employ an LSTM network to model the evolution of latent space for addressing the macro-drift. However, using an LSTM to model the evolution of latent space has been significantly explored in the past. I suggest authors to refer to the paper (and its citations): Deep State Space Models for Time Series Forecasting

The empirical success of LSTM-based evolution is well-known, hence I believe that the solution to the macro-drift issue is short of novelty. The proposed solution can act as a strong baseline for a more novel solution. I suggest authors explore more aggressive approaches for extrapolation than only using a vanilla LSTM for predicting the evolution of parameters.

2. More details on the solution proposed for modeling micro-drifts are needed. Perhaps a more detailed description of the relation network (R) and embedding function (g) is required. What are the alternative ways to implement them and why proposed implementations work the best should also be addressed. I also suspect any off-the-shelf MLP used for the relation network is at the risk of overfitting. I would like to know authors' comments on that, whether they saw any overfitting isses? If possible, please provide the necessary ablations.

**Questions:**

1. In Figure 1a, it is not clear which dataset the shown time-series corresponds to. If this is a real-world dataset, please provide the reference.
2. Are there any external features and time-related features used in the models? If so, how does their presence affect the overall meta-learning process? Knowing this can also help in evaluating the quality of the proposed framework against external signals which are much easier to learn from.

---

> ### Author Response · Authors · 2023-11-22
> **Responce to Reviewer 9YY8**
>
> Thanks for your constructive suggestions! We address your concerns in the following.
>
> **{Q1}** The novelty of using an LSTM to model the evolution of latent space, e.g., DeepState; explore more aggressive approaches for extrapolation than only using a vanilla LSTM for predicting the evolution of parameters
>
> **{R1}** While there exist various works, such as DeepState, that model the evolution of time series with hidden states, LEAF takes an innovative approach by representing the model parameters with evolving hidden states. This addresses the challenge of the model changing as data experiences concept drift. Our primary objective in this research is to address concept drift in online time series forecasting using a general framework. To achieve this, we have simplified the design of the network architecture to ensure that any improvements are solely achieved through the learning framework, rather than relying on a complex network structure. Similarly, other networks in LEAF, such as the loss network $s(\cdot)$, embedding network $g(\cdot)$ and relation network ${R}(\cdot)$, are also kept simple, with just two layers of Multilayer Perceptron (MLP). However, it is possible to substitute them with more sophisticated networks to achieve better performance in practical applications. Nevertheless, in this research work, our focus is solely on investigating the framework itself.
>
> **{Q2}** Detailed description of the relation network (R) and embedding function (g) and the alternative ways to implement them and why proposed implementations work the best should also be addressed; Overfitting problem of off-the-shelf MLP.
>
> **{R2}** Actually, in our implementation, both the relation network (R) and embedding function (g) are implemented as 2-layer MLPs. The reason for this design choice is explained in **{R1}**. Adjusting the model for each individual sample may potentially lead to overfitting problems. To mitigate this issue, we have introduced several strategies. Firstly, we optimize the sample-specific model in a low-dimensional latent space, which limits the degree of freedom for adjusting the model. This can be seen as an advanced and more general version of adjusting the last linear layer with a few parameters for each individual sample [1]. Additionally, we only perform one iteration of SGD with a learning rate of 1e-3 to adjust the sample-specific latent embeddings. As a result, the adjusted embedding will not deviate significantly from the base embedding, thereby avoiding overfitting.
>
> [1] Gerald Woo, et al. Learning Deep Time-index Models for Time Series Forecasting. ICML 2023.
>
> **{Q3}** Which data is shown in Fig.1(a)?
>
> **{R3}** The data shown in Fig.1(a) is actually synthetic data. We chose to use synthetic data because when illustrating the macro-drift of real-world data over a significant period of time, the micro-drift that occurs over a shorter duration may not be easily discernible. However, we acknowledge that using real-world data would be more representative and plan to replace the synthetic data with real data in the final version of our work.
>
> **{Q4}** Are there any external features and time-related features used in the models? If so, how does their presence affect the overall meta-learning process?
>
> **{R3}** We have introduced the time-related features (e.g., hour of day, day of week, and month of year) in LEAF and other baselines. Besides, these covariates are also used in the target prediction model. This step, which is a standard practice in time series forecasting, has been omitted in the paper for brevity. These time-related features are embedded with embedding layers which are then added to the time series representations. These features provide information of global time stamp, helping the model understand the overall evolving pattern of time series.

---

> ### Author Response · Authors · 2023-11-23
>
> Dear Reviewer 9YY8,
>
> Since the End of author/reviewer rebuttal is approaching, may we know if our response addresses your main concerns? If so, we kindly ask for your reconsideration of the score. Should you have any further advice on the paper and/or our rebuttal, please let us know and we will be more than happy to engage in more discussion and paper improvements.
>
> Thank you so much for devoting time to improving our work!

---

### Meta-Review · Area_Chair_kvxd · 2023-12-07

**Metareview:**

The paper proposes a two-stage meta-learning framework, LEAF (Learning to Extrapolate and Adjust for Forecasting), addressing macro and micro-drifts in time-series forecasting. It uses an LSTM for macro-drifts and a meta-learning framework for micro-drifts, allowing online learning.

Strengths include its clear presentation, practical relevance for online learning, and essential benchmarking. Weaknesses involve lack of novelty in addressing macro-drifts with LSTM, insufficient detail on micro-drift solution, missing theoretical foundation, and unclear network architecture.

There's a call for more baselines, efficiency comparisons, and addressing space-time complexity. Some reviewers requested clearer explanations of the surrogate loss and the adjustment process for each sample, while suggesting a need for simplification in the paper and a baseline comparison with previous period models.

**Justification For Why Not Higher Score:**

Reviewers appreciated the paper's structure and methodology but had concerns about a few areas. There's acknowledgment of the paper's clear writing and the importance of addressing concept drift in time series forecasting. However, some reviewers suggested areas of improvement, such as the need for more novelty in addressing macro-drift, more details on the solution for micro-drifts, theoretical support, and comparisons with additional baselines.

There's a consensus that the paper has strengths in its approach but also lacks clarity in certain aspects like theoretical foundation, details on network architecture, and computational complexities. Reviewers also pointed out the necessity for clearer explanations regarding specific methodologies, potential efficiency issues, and comparisons with other baseline models.

Addressing these concerns could potentially elevate the paper's score. Improving the novelty in addressing macro-drift, providing more details on micro-drift solutions, offering theoretical support, enhancing clarity on methodologies, and considering additional comparisons with relevant baselines might lead to a higher rating from the reviewers.

**Justification For Why Not Lower Score:**

N/A

---

### Decision · Program_Chairs · 2024-01-16

Reject